# Fast Byte Latent Transformer

Julie Kallini [1 2]   Artidoro Pagnoni [1]   Tomasz Limisiewicz [1 3]   Gargi Ghosh [1]   Luke Zettlemoyer [1 3]
Christopher Potts [2]   Xiaochuang Han [1]   Srinivasan Iyer [1]

## Abstract

Recent byte-level language models (LMs) match the performance of token-level models without relying on subword vocabularies, yet their utility is limited by slow, byte-by-byte autoregressive generation. We address this bottleneck in the Byte Latent Transformer (BLT) through new training and generation techniques. First, we introduce **BLT Diffusion (BLT-D)**, a new model and our fastest BLT variant, trained with an auxiliary block-wise diffusion objective alongside the standard next-byte prediction loss. This enables an inference procedure that generates multiple bytes in parallel per decoding step, substantially reducing the number of forward passes required to generate a sequence. Second, we propose two extensions inspired by speculative decoding that trade some of this speed for higher generation quality: **BLT Self-speculation (BLT-S)**, in which BLT's local decoder continues generating past its normal patch boundaries to draft bytes, which are then verified with a single full-model forward pass; and **BLT Diffusion+Verification (BLT-DV)**, which augments BLT-D with an autoregressive verification step after diffusion-based generation. All methods may achieve an estimated memory-bandwidth cost over 50% lower than BLT on generation tasks. Each approach offers its own unique advantages, together removing key barriers to the practical use of byte-level LMs.

## 1. Introduction

Byte-level (also known as *tokenizer-free*) language models operate directly on raw bytes rather than a predefined vocabulary of tokens. By avoiding subword tokenization,

they address several well-known shortcomings of token-level models, including sensitivity to input noise (Pruthi et al., 2019; Sun et al., 2020), handling structured or out-of-domain inputs (Dagan et al., 2024; Singh & Strouse, 2024; Zhou et al., 2024), limited character-level understanding (Kaushal & Mahowald, 2022; Huang et al., 2023; Edman et al., 2024), and multilingual disparities (Ahia et al., 2023; Petrov et al., 2023; Liang et al., 2023). However, byte-level models have historically suffered from poor efficiency due to longer input and output sequences (Xue et al., 2022).

Recent architectural innovations have substantially narrowed this efficiency gap. Modern byte-level models often group bytes into larger units, use hierarchical computation, or replace full attention with more efficient sequence modeling mechanisms (El Boukkouri et al., 2020; Clark et al., 2022; Tay et al., 2022; Nawrot et al., 2022; 2023; Yu et al., 2023; Slagle, 2024; Wang et al., 2024; Kallini et al., 2025; Zheng et al., 2025; Pagnoni et al., 2025; Hwang et al., 2025). For example, the **Byte Latent Transformer** (BLT; Pagnoni et al., 2025) dynamically groups bytes into variable-length *patches* based on input complexity. Its hierarchical design concentrates computation on latent token representations, allocating more compute to complex patches of text and yielding better scaling behavior than token-level models.

These advances reduce the *compute* cost of byte-level models, but inference still faces a *memory bandwidth* bottleneck. In modern LLM inference, generation cost is often dominated by repeatedly loading model weights and accessing key-value caches (Pope et al., 2023; Kwon et al., 2023; Yuan et al., 2024). Even when most computation is performed over latent token representations, standard byte-level decoding still generates one byte at a time. Since a typical subword token corresponds to several bytes, an autoregressive byte-level model such as BLT requires multiple decoder forward passes to generate the same amount of text represented by a single subword token. This paper targets that bottleneck. Our goal is to enable byte-level parallel generation while preserving the main benefits of BLT: operating directly on bytes, using dynamic patching, and concentrating computation in latent token representations.

We first draw inspiration from diffusion language models (dLMs), which improve decoding efficiency by generating

[1]FAIR at Meta [2]Stanford University [3]University of Washington. Correspondence to: Julie Kallini <kallini@stanford.edu>, Srinivasan Iyer <sviyer@meta.com>.

*Proceedings of the 43rd International Conference on Machine Learning*, Seoul, South Korea. PMLR 306, 2026. Copyright 2026 by the author(s).

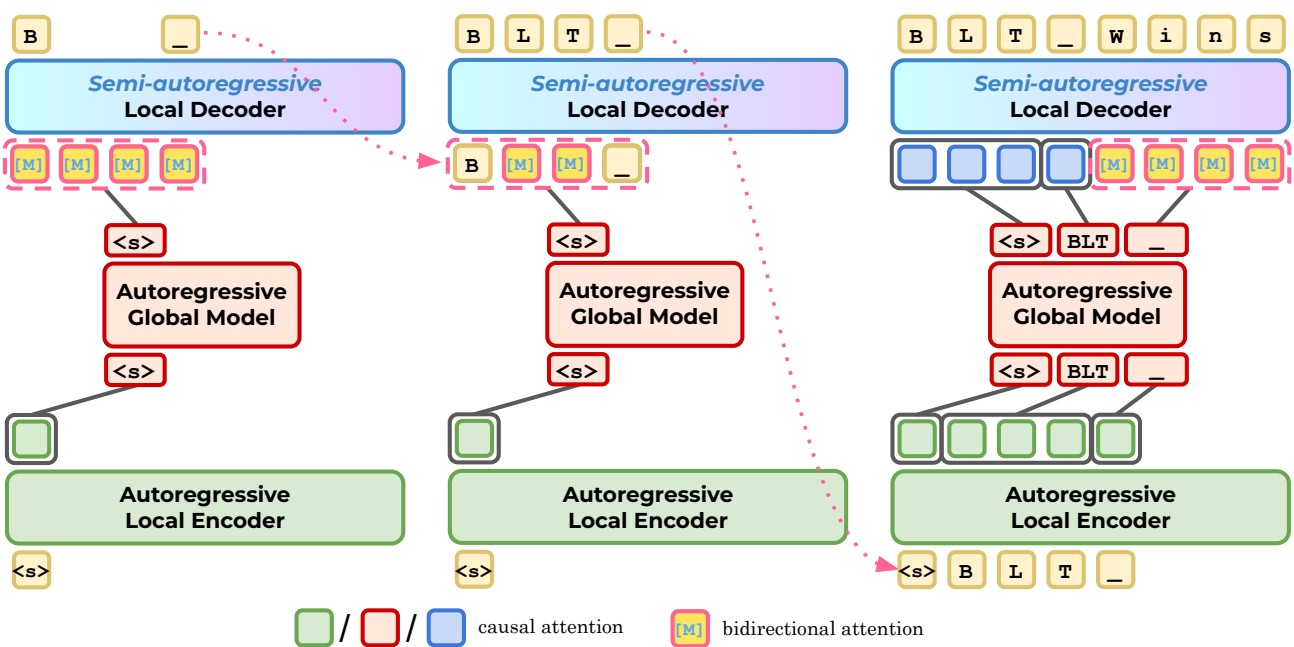

*Figure 1.* **BLT-D inference.** The encoder creates latent token representations from variable-length patches of bytes. The large global model predicts the next latent token. The decoder initializes a fixed-length block of [MASK] tokens and generates bytes in parallel via semi-autoregressive text diffusion, conditioning on the last latent token. **Compared to BLT, this inference approach decreases the forward passes/network function evaluations (NFEs) of all model components (encoder, global model, and decoder).**

multiple tokens in parallel within a single forward pass (Sahoo et al., 2024; Lou et al., 2024; Wu et al., 2025; Nie et al., 2025; Arriola et al., 2025), reducing memory bandwidth per generated byte. However, existing text diffusion methods are not directly designed for byte-level architectures whose latent tokens are constructed dynamically from variable-length patches.

We introduce **BLT Diffusion (BLT-D)** (Figure 1), a new byte-level model that combines BLT's hierarchical latent tokenization with block-wise discrete diffusion. BLT-D retains BLT's local encoder and global model structure, but modifies training and decoding so that the local decoder can generate a fixed-size block of future bytes in parallel. During training, BLT-D's decoder receives both a clean byte sequence and a corrupted sequence of fixed-length byte blocks. These blocks are constructed from dynamically segmented patches but can extend beyond individual patch boundaries, allowing the decoder to learn to predict future bytes beyond the average BLT patch size. The decoder is trained with a combined objective: the standard autoregressive next-byte prediction loss on clean bytes, and a masked-byte prediction loss on corrupted byte blocks. At inference time, BLT-D initializes a block of masked byte positions and iteratively unmasks multiple positions per decoder step, conditioning on the most recent latent representation. This reduces the number of required decoder, encoder, and global model evaluations per generated sequence.

Next, we introduce two inference extensions to BLT and BLT-D that draw inspiration from speculative decoding (Leviathan et al., 2023; Zhang et al., 2024; Cai et al., 2024). Unlike prior speculative decoding methods that typically use a separate draft model or additional speculative layers, our methods exploit the existing hierarchical structure of BLT and BLT-D. The first extension is **BLT Self-speculation (BLT-S)**, in which standard BLT uses *only its decoder* to draft bytes beyond its usual patch boundaries, then verifies its drafts with a forward pass through the full model. While this method increases decoder calls, it minimizes overall model calls compared to BLT. The second extension is **BLT Diffusion+Verification (BLT-DV)**; since BLT-D is also trained on a next-byte prediction objective, it can also verify the bytes it has drafted with diffusion using next-byte prediction, improving generation quality. Both methods guarantee identical outputs to standard autoregressive decoding with dynamic patching and latent tokenization.

Across our experiments, BLT-D is our fastest model and inference method, achieving over 50% lower estimated memory-bandwidth cost compared to BLT on translation and code generation tasks. With larger diffusion block sizes, BLT-D may achieve up to 92% reduction, with some degradation in task performance. BLT-DV recovers some of this performance while still achieving up to 81% reduction compared to BLT, and BLT-S achieves up to 77% reduction with no loss in task performance. Overall, each of these

methods has its own unique advantages and helps to further close the inference efficiency gap between byte-level and subword-level models.

**Conflict of Interest Disclosure.** The authors JK, AP, TL, GG, LZ, XH, and SI were employed by FAIR at Meta at the time of this paper's submission. Meta leads the development of BLT models.

## 2. Background and Related Work

In this section, we provide background on BLT and diffusion language models. We further discuss speculative decoding in Section 5, where we introduce our extensions.

### 2.1. Byte Latent Transformer

BLT is a byte-level architecture that operates directly on raw byte sequences while matching the performance of subword tokenization-based language models at scale. BLT dynamically groups bytes into variable-length *patches*, which serve as the primary units of computation. Patches are constructed using an entropy-based segmentation strategy driven by next-byte uncertainty estimated by a small auxiliary byte-level language model. Given a byte input sequence $x = [x_1; x_2; \ldots; x_N] \in \mathcal{V}^N$ of length $N$, where $\mathcal{V}$ is a small byte vocabulary, the sequence is split into $M \approx \frac{N}{4}$ variable-length patches $[p_1; p_2; \ldots; p_M]$. High-entropy regions are segmented into shorter patches, while more predictable spans are grouped into longer patches, thus controlling how frequently the global model is invoked.

**Architecture overview.** BLT's architecture creates latent token representations that mix byte- and patch-level information. It consists of three components: a local encoder $\mathcal{E}$, a global transformer $\mathcal{G}$, and a local decoder $\mathcal{D}$. The local encoder embeds the length-$N$ byte input $x$ to create initial byte representations $\mathbf{X} = [\mathbf{x}_1; \mathbf{x}_2; \ldots; \mathbf{x}_N] \in \mathbb{R}^{N \times d_{\text{local}}}$, where $d_{\text{local}}$ is the hidden dimensionality of the local encoder and decoder modules and where $\mathbf{x}_i$ is the embedding of byte $x_i$. The encoder then processes $\mathbf{X}$ into $M$ latent token representations $\mathbf{T} = [\mathbf{t}_1; \mathbf{t}_2; \ldots; \mathbf{t}_M] \in \mathbb{R}^{M \times d_{\text{global}}}$, where $d_{\text{global}}$ is the hidden dimensionality of the global model. The global Transformer then maps $\mathbf{T}$ to output latent token representations $\mathbf{O} = [\mathbf{o}_1; \mathbf{o}_2; \ldots; \mathbf{o}_M] \in \mathbb{R}^{M \times d_{\text{global}}}$. Since our method modifies the decoder, we omit further details of $\mathcal{E}$ and $\mathcal{G}$ and refer the reader to Pagnoni et al. 2025.

**Local decoder.** The local decoder $\mathcal{D}$ autoregressively decodes the final latent token representations $\mathbf{o}$ into a sequence of output bytes $y = [y_1; y_2; \ldots; y_N] \in \mathcal{V}^N$ using $L_{\mathcal{D}}$ lightweight Transformer layers. At each layer, byte-level hidden states are updated via cross-attention to latent token representations before applying a standard Transformer

layer. Let $\mathbf{D}_l = [\mathbf{d}_{l,1}; \mathbf{d}_{l,2}; \ldots; \mathbf{d}_{l,N}] \in \mathbb{R}^{N \times d_{\text{local}}}$ denote the byte hidden states of a length-$N$ byte sequence output by layer $l$ of the decoder, with $\mathbf{D}_0 \in \mathbb{R}^{N \times d_{\text{local}}}$ being the initial representations from an embedding lookup for $y$. For each decoder layer $l \in \{1, \ldots, L_{\mathcal{D}}\}$, the cross-attention from byte hidden states to latent token representations is computed as

$$\mathbf{B}_l = \mathbf{D}_{l-1} + \mathbf{W}_o \left( \text{softmax}\left( \frac{\mathbf{Q}\mathbf{K}^\top}{\sqrt{d_k}} \right) \mathbf{V} \right), \quad (1)$$

where $\mathbf{Q}_i = \mathbf{W}_q(\mathbf{d}_{l-1,i})$, $\mathbf{K}_j = \mathbf{W}_k(D_C(\mathbf{o}_j))$, and $\mathbf{V}_j = \mathbf{W}_v(D_C(\mathbf{o}_j))$. Here, $d_k$ is the dimensionality of the key vectors for a single attention head. $\mathbf{W}_q$, $\mathbf{W}_k$, and $\mathbf{W}_v$ are the query, key, and value projection matrices, $D_C(\cdot)$ denotes a linear transformation and splitting function applied to latent token representations, and $\mathbf{W}_o$ is the output projection. The cross-attention does not use positional encodings. The updated byte representations are then produced by

$$\mathbf{D}_l = \text{DecoderTransformerLayer}(\mathbf{B}_l). \quad (2)$$

The decoder Transformer layer employs multi-head attention, pre-LayerNorm, and RoPE positional encodings.

### 2.2. Diffusion Language Models

Diffusion models define generative distributions by progressively corrupting data through a forward noising process and learning a reverse process that iteratively removes noise. Recent work extends this framework to discrete domains such as text by defining stochastic corruption processes over token sequences, enabling training of diffusion language models (dLMs) with diffusion-style objectives and generation over discrete tokens (Austin et al., 2021a; Campbell et al., 2022; Li et al., 2022; Gulrajani & Hashimoto, 2023; Lou et al., 2024). These models are typically non-autoregressive, employing bidirectional attention over all tokens, or semi-autoregressive, using bidirectional attention within fixed-length blocks while maintaining causal dependencies across blocks (Arriola et al., 2025; Gat et al., 2025). Here, we focus on absorbing discrete diffusion with conventions similar to those presented by Ye et al. (2025) and Nie et al. (2025), which is conceptually very similar to masked language models (Devlin et al., 2019).

**Absorbing Discrete Diffusion.** We draw a clean text sequence $x^0 = [x_1^0; x_2^0 \ldots; x_N^0] \in \mathcal{V}^N$ from the data distribution, where $\mathcal{V}$ is the vocabulary and $N$ is the sequence length. We define a discrete diffusion process based on random input masking: given $x^0$, we sample a continuous diffusion timestep (noise level) $t \sim \mathcal{U}(0, 1)$ and independently replace each position with a special [MASK] token with probability $t$, producing a corrupted sequence $x^t$. The forward corruption distribution $q$ is

$$q(x_i^t = \text{[MASK]} \mid x_i^0) = t, \quad q(x_i^t = x_i^0 \mid x_i^0) = 1 - t, \quad (3)$$

with independence across positions. Prior work has shown that this masking process can be interpreted as the marginal of a discrete diffusion model with an absorbing state, where [MASK] is absorbing and $t$ controls the diffusion time.

We parameterize a denoising model $p_\theta(x_i^0 \mid x^t, t)$ that predicts the original token values at masked positions, conditioned on the partially observed sequence and the noise level. Training minimizes the weighted denoising objective

$$\mathcal{L}(\theta) = -\mathbb{E}_{x^0, t, x^t} \left[ \frac{1}{t} \sum_{i=1}^N \mathbb{1}_{[x_i^t=[\text{MASK}]]} \log p_\theta(x_i^0 \mid x^t, t) \right], \tag{4}$$

which has been shown to correspond to a simplified evidence lower bound (ELBO) on the data log-likelihood, or equivalently, an upper bound on the negative log-likelihood (Shi et al., 2024; Gong et al., 2025). Following Ye et al. (2025) and Nie et al. (2025), we do not embed the timestep $t$ into the architecture directly and instead assume that it is implicitly encoded through the input data corruption.

## 3. BLT Diffusion

BLT achieves scalable and efficient byte-level modeling by dynamically allocating compute resources through hierarchical latent tokenization. However, inference speed remains a significant bottleneck, as traditional autoregressive generation proceeds one byte at a time. BLT-D directly addresses this challenge by introducing block diffusion decoding in a way that is fully compatible with BLT's hierarchical architecture, reducing model calls and therefore memory bandwidth at inference. We adapt the absorbing diffusion framework from Section 2.2 to operate over fixed-size blocks within BLT's decoder.

### 3.1. BLT-D Inference

BLT-D inference decodes a fully masked block in parallel in much fewer iterations than autoregressively generating a byte at a time (Figure 1). BLT-D's encoder $\mathcal{E}$ and global model $\mathcal{G}$ operate exactly like BLT, as described in Section 2.1. Given a length-$N$ prefix $x = [x_1; \ldots; x_N] \in \mathcal{V}^N$, the patcher segments $x$ into $M$ variable-length patches. The encoder $\mathcal{E}$ produces byte embeddings $\mathbf{X} \in \mathbb{R}^{N \times d_{\text{local}}}$ and encodes them into latent token representations $\mathbf{T} = [\mathbf{t}_1; \ldots; \mathbf{t}_M] \in \mathbb{R}^{M \times d_{\text{global}}}$. The global model $\mathcal{G}$ outputs contextual latent tokens $\mathbf{O} = [\mathbf{o}_1; \ldots; \mathbf{o}_M] \in \mathbb{R}^{M \times d_{\text{global}}}$.

For block diffusion inference, the decoder $\mathcal{D}$ receives as input both the latent token representations $\mathbf{O}$ and a byte sequence $x' = [x_1; \ldots; x_N; x_{N+1}; \ldots; x_{N+B}] \in \mathcal{V}^{N+B}$, where $[x_{N+1}; \ldots; x_{N+B}] = \{[\text{MASK}]\}^B$ form a block of $B$ masked positions. $\mathcal{D}$ iteratively computes forward passes over $x'$ until the entire block of $B$ bytes is unmasked. See Algorithm 1 for a more detailed description of the generation procedure.

**Attention Patterns.** Let $i \in \{1, \ldots, N + B\}$ index positions in $x'$. Let $p(i)$ denote the patch index for position $i$ in $x'$. For the decoder's cross-attention module, for clean positions in the sequence ($i \leq N$), each position attends to the latent token $\mathbf{o}_{p(i)-1}$ corresponding to the previous patch, except for the final byte of each patch, which attends to its own latent token $\mathbf{o}_{p(i)}$ (consistent with BLT). For positions in the masked block ($i > N$), all positions attend to the last latent token $\mathbf{o}_M$. For $\mathcal{D}$'s self-attention, the attention mask $A \in \{0, 1\}^{(N+B) \times (N+B)}$ is defined as follows. For prefix positions ($i \leq N$), $\mathcal{D}$'s self-attention is causal: $A_{ij} = 1$ if $j \leq i$. For block positions ($i > N$), self-attention is fully bidirectional: $A_{ij} = 1$ for all $j \leq N + B$. We provide a visualization of these inference attention masks in Section B.

**Block Unmasking Strategy.** The choice of which bytes to unmask at each decoder forward pass affects both the generation quality and the degree of parallelism. We consider two unmasking strategies that differ in how they select masked positions for decoding: *confidence-based thresholding*, where a position is unmasked if its top-predicted probability exceeds a threshold $\alpha$, or *entropy-bounded sampling*, where positions are unmasked if their cumulative entropy is below a threshold $\gamma$. This process repeats over decoder steps, updating the set of masked positions, until all masks have been resolved. We provide details of these strategies in Section C, and an analysis of generations in Section H.

**Speedup.** Compared to standard autoregressive decoding, this approach reduces the number of decoder forward passes: generating a block of size $B$ requires $s$ unmasking steps rather than $B$ sequential steps. Usually, $s < B$, which results in a speedup. Additionally, the encoder and global model are invoked less frequently, as these components are called once per block—typically larger than the average patch—rather than at every new patch. Furthermore, the clean prefix and the first $M - 1$ latent tokens from $\mathcal{E}$, $\mathcal{G}$, and $\mathcal{D}$ can be cached, with only the final latent token and drafted block requiring recomputation.

### 3.2. BLT-D Training

BLT-D uses a new training method that enables byte diffusion decoding over latent tokens using specific training data preprocessing, special attention masking in its decoder, and a new loss function. These additions enable BLT-D to predict diffusion blocks that span future bytes far beyond BLT's typical patch size.

**Training Data Preprocessing.** Given an input byte sequence $x = [x_1; x_2; \ldots; x_N] \in \mathcal{V}^N$, we segment $x$ into $M$ variable-length patches, each starting at index $s_i$ for

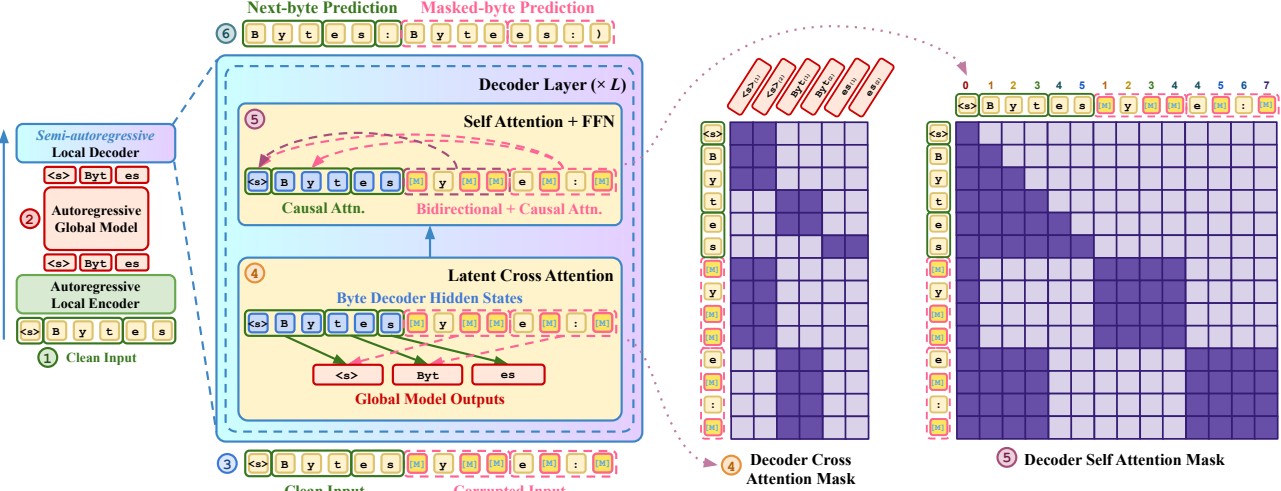

*Figure 2.* **BLT-D training forward pass.** (1-2) The encoder and global model process the clean input. (3) Clean and corrupted inputs are concatenated and passed to the decoder. (4) Byte hidden states cross-attend to their corresponding latent representations from the global model. (5) For the clean portion, self-attention is causal; for the corrupted portion, self-attention is bidirectional within each block, and causal towards previous clean patches. (6) Next-byte prediction loss is computed for the clean sequence, and masked byte prediction/diffusion loss is computed for the corrupted sequence.

$i \in \{1, \ldots, M\}$. For each patch $p_i$ for $i \geq 2$, we construct a block $b_{i-1} = [x_{s_i}; x_{s_i+1}; \ldots; x_{s_i+B-1}] \in \mathcal{V}^B$, resulting in $M - 1$ fixed-length blocks that have a correspondence to the patches, but may contain bytes from future patches. All blocks are concatenated to form $x_{\text{block}} = [b_1; b_2; \ldots; b_{M-1}] \in \mathcal{V}^{B \cdot (M-1)}$, and we record the original byte positions for each block to use for positional encodings. To corrupt the data, we sample $t \sim \mathcal{U}(0, 1)$ and independently replace each byte in $x_{\text{block}}$ with a [MASK] token with probability $t$, yielding the masked input $x_{\text{block}}^t = [b_1^t; b_2^t; \ldots; b_{M-1}^t] \in \{\mathcal{V} \cup [\text{MASK}]\}^{B \cdot (M-1)}$. The model is trained to reconstruct the clean bytes $x_{\text{block}}$ from the corrupted blocks $x_{\text{block}}^t$. Both the block construction and diffusion corruption enable BLT-D to predict bytes beyond its average patch size. See Figure 6 for a visualization.

**Decoder Architecture and Attention Patterns.** The primary architectural innovation in BLT-D lies in the local decoder $\mathcal{D}$, which enables block diffusion decoding. A detailed visualization of the architecture during a training forward pass is shown in Figure 2. BLT-D initializes the decoder input $\mathbf{D}_0$ from embeddings of the concatenated clean and corrupted sequences: $\mathbf{D}_0 = \text{Embed}([x; x_{\text{block}}^t])$.

For each byte in $\mathbf{D}_0$, cross-attention is applied to the corresponding output latent token in $\mathbf{O}$. Clean sequence positions associated with patch $p_i$ cross-attend to the previous latent token $\mathbf{o}_{i-1}$, except final bytes, which attend to their own latent token $\mathbf{o}_i$, consistent with BLT. Corrupted sequence positions associated with patch $p_i$ cross-attend to the previous latent token $\mathbf{o}_{i-1}$. This pattern maintains the alignment between patches and blocks throughout the sequence. Self-

attention in $\mathcal{D}$ uses a causal mask for the clean sequence and bidirectional attention within each block of the corrupted sequence. Each byte within a given block in the corrupted sequence also attends causally to all previous clean bytes. RoPE positional encoding uses the original positional indices as we defined previously.

**Loss Function.** We use a loss function that combines next-byte prediction on the clean sequence with masked reconstruction on the corrupted sequence. First, recall the clean sequence $x = [x_1; \ldots; x_N] \in \mathcal{V}^N$, segmented into $M$ patches with starting indices $s_i$. We compute an autoregressive next-byte prediction loss:

$$\mathcal{L}_{\text{clean}}(\theta) = -\sum_{i=1}^{N} \log p_\theta(x_i \mid x_{<i}) \quad (5)$$

Here, $p_\theta(x_i \mid x_{<i})$ denotes the model's predicted probability of byte $x_i$ given the prefix $x_{<i}$. Next, recall the corrupted sequence $x_{\text{block}}^t = [b_1^t; \ldots; b_{M-1}^t]$. Each corrupted block $b_{i-1}^t = [x_{s_i}^t; x_{s_i+1}^t; \ldots; x_{s_i+B-1}^t] \in \{\mathcal{V} \cup [\text{MASK}]\}^B$ for $i \in \{2, \ldots, M\}$, with each byte masked with probability $t$. For each corrupted block $b_{i-1}^t$, let $b_{i-1,k}^t$ denote the $k$-th byte of the block. The masked diffusion loss is:

$$\mathcal{L}_{\text{mask}}(\theta) =$$
$$-\frac{1}{t} \sum_{i=2}^{M} \sum_{k=0}^{B-1} \mathbb{1}_{[b_{i-1,k}^t = [\text{MASK}]]} \log p_\theta(x_{s_i+k} \mid b_{i-1}^t, x_{<s_i})$$

$$(6)$$

where $\mathbb{1}_{[b_{i-1,k}^t = [\text{MASK}]]}$ is an indicator function that is 1 if the $k$-th byte of block $b_{i-1}^t$ is masked, and 0 otherwise. The

model reconstructs the clean byte $x_{s_i+k}$, conditioned on the partially masked block and the clean prefix preceding the block, consistent with the self-attention masking pattern described above. The scaling by $1/t$ follows the absorbing discrete diffusion loss discussed previously in Section 2.2.

The total training loss is the sum of the clean sequence loss and the masked diffusion loss:

$$\mathcal{L}_{\text{total}}(\theta) = \mathcal{L}_{\text{clean}}(\theta) + \mathcal{L}_{\text{mask}}(\theta) \tag{7}$$

This combined objective encourages the model to learn both autoregressive next-byte prediction and robust reconstruction of masked bytes in block-wise corrupted sequences.

## 4. Pre-training and Generation Experiments

In this section, we detail the architectures and hyperparameters of each BLT and BLT-D model we train, as well as the pre-training dataset and optimization settings. We evaluate our models on four generation tasks and discuss the efficiency metrics and results.

### 4.1. Models, Pre-training Data, and Optimization

We pre-train four model types: one BLT and three BLT-D variants with block sizes of 4, 8, and 16, referred to as BLT-D-4, BLT-D-8, and BLT-D-16, respectively. For each model type, we train both 1B- and 3B-parameter versions. Our 1B BLT and BLT-D models consist of a global model with 1.28 billion parameters, a local encoder with 19 million parameters, and a local decoder with 160 million parameters. Our 3B BLT and BLT-D models include a global model with 2.82 billion parameters, a local encoder with 26 million parameters, and a local decoder with 160 million parameters. All models employ entropy patching, using an average patch size of 4 bytes and a maximum patch size of 8 bytes. To ensure comparability, all models are trained on the BLT-1T dataset from Pagnoni et al. 2025, which consists of 1 trillion tokens collected from various public sources and includes a subset of the pre-training data released by Datacomp-LM (Li et al., 2024). For additional details on model implementation, hyperparameters, and pre-training optimization settings, see Section E.

### 4.2. Generation Tasks, Settings, and Metrics

We evaluate our BLT and BLT-D models on four generation tasks: two translation tasks and two coding tasks. For translation, we evaluate French-to-English and German-to-English (4-shot) using the FLORES-101 benchmark (Goyal et al., 2022), with performance measured by SentencePiece BLEU. For coding, we assess models on HumanEval (0-shot) (Chen et al., 2021) and MBPP (3-shot) (Austin et al., 2021b), reporting pass@1 scores. All task-evaluation inference uses greedy decoding. For BLT-D models, we experi-

ment with both confidence-based unmasking and entropy-bounded sampling as diffusion unmasking strategies, conducting hyperparameter sweeps for each.

Efficiency is evaluated using three metrics: (1) the average number of decoder network function evaluations (NFEs, or forward passes) per output sequence; (2) the average number of encoder/global model NFEs per output sequence; and (3) an estimate of the memory bandwidth required for parameter memory loads during evaluation. The total memory bandwidth, measured in gigabytes, is calculated as follows:

$$\frac{b\left[N_{\text{dec}} \cdot P_{\text{dec}} + N_{\text{enc}} \cdot (P_{\text{enc}} + P_{\text{glob}})\right]}{10^9} \tag{8}$$

Here, $N_{\text{dec}}$ and $N_{\text{enc}}$ represent the average number of function evaluations for the decoder and encoder/global model, respectively. $P_{\text{dec}}$, $P_{\text{enc}}$, and $P_{\text{glob}}$ denote the number of parameters in the decoder, encoder, and global model. The variable $b$ specifies the number of bytes required to represent each parameter; in our calculations, we set $b = 2$ to reflect 16-bit precision. This formulation assumes that evaluations are performed with a small KV cache and batch size, so the memory bandwidth is dominated by loading model weights. Small batch sizes are common in local serving and latency-oriented applications, where execution speed is prioritized over batching efficiency. BLT-D supports KV caching, and therefore benefits from any techniques that reduce KV-cache memory footprint. Alternatively, memory bandwidth may be interpreted as a weighted function of NFEs for each model component.

### 4.3. Generation Task Results

We present the performance of our 3B models across a range of generation tasks, as illustrated in Figure 3. For clarity and brevity, this section focuses on representative BLT-D models utilizing confidence-based unmasking as the diffusion generation strategy, with a confidence threshold of $\alpha = 0.7$. For comprehensive results for both 1B and 3B model variants, including full inference hyperparameter sweeps using both confidence-based unmasking and EB sampling strategies, please refer to Section F and Section G. We additionally evaluate longer document-level translation for the 3B models in Section G.

Across all evaluated tasks, BLT-D models consistently outperform BLT in terms of efficiency. Specifically, BLT-D variants achieve substantial reductions in both decoder NFEs and encoder/global NFEs, resulting in large memory bandwidth decreases. For example, BLT-D-4 nearly matches BLT's task scores while requiring less than half the NFEs and memory bandwidth. Both BLT-D-4 and BLT-D-8 demonstrate strong task performance with great gains in efficiency, especially on the translation tasks.

Increasing the block size in BLT-D models (e.g., BLT-D-16)

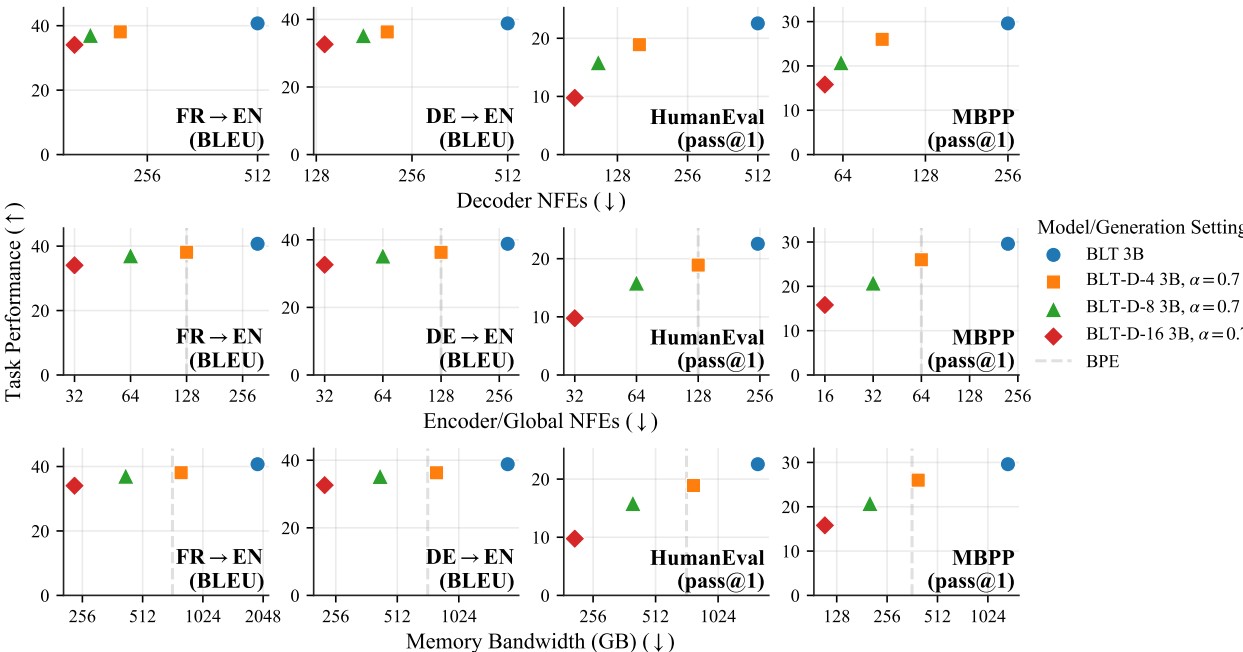

*Figure 3.* Generation task results of 3B-parameter variants of BLT, BLT-D-4, BLT-D-8, and BLT-D-16. Higher is better for task performance; lower is better for NFEs and memory bandwidth. The NFEs and memory bandwidth for a byte-pair encoding (BPE) model matching BLT's global model size are shown as a dashed line. BLT-D models are substantially faster than BLT while maintaining strong task performance, especially for translation. BLT-D-16 offers the most efficiency, with reduced performance on the coding-related tasks.

leads to even greater reductions in NFEs, highlighting the scalability of this approach. BLT-D-16 achieves an 87–92% reduction in memory bandwidth compared to BLT, making it the fastest model in our evaluations. However, while BLT-D-16 remains competitive on translation tasks, its enhanced efficiency comes at the expense of lower performance on coding-related tasks. This suggests a trade-off between speed and generation quality as block size increases. These results highlight the versatility of BLT-D models, enabling fast generation while allowing flexibility to adjust the block size to suit specific application needs. Section I further studies parameter scaling by training additional 400M-parameter BLT and BLT-D models under the same 1T-token budget; across 400M, 1B, and 3B scales, all model families improve, and on translation tasks the gap between BLT and the diffusion-based variants narrows as model size increases.

## 5. Extensions: BLT-S and BLT-DV

Based on our observations from the previous section, BLT achieves strong task performance but suffers from slow generation, while BLT-D greatly improves efficiency but can lose quality at larger block sizes. To improve both models, we draw inspiration from speculative decoding, which accelerates autoregressive generation by separating decoding into a fast *drafting* stage and a slower *verification* stage (Leviathan et al., 2023). In standard speculative de-

coding, a lightweight *draft model* proposes multiple future tokens, and the large *target model* verifies those proposals in parallel, accepting a prefix of the draft while preserving the target model's output distribution. Subsequent work has reduced the need for a separate draft model by using self-speculation or additional speculative heads (Zhang et al., 2024; Cai et al., 2024).

Our setting is different: BLT and BLT-D already decompose generation into lightweight byte-level decoding and more expensive encoder/global-model computation. We therefore use the existing model components themselves as drafters: BLT-S drafts with BLT's local decoder beyond normal patch boundaries, while BLT-DV drafts with BLT-D's diffusion decoder and verifies with autoregressive next-byte prediction. These inference extensions require no architectural changes or additional training.

### 5.1. BLT Self-speculation

We introduce a new approach to enhance BLT's inference efficiency by enabling its decoder to speculate beyond where it would normally segment patches. In standard BLT inference, the entropy-based patcher halts generation whenever a high-entropy byte is produced, prompting a new invocation of the encoder and compute-intensive global model. This patching typically occurs every four bytes. Instead of

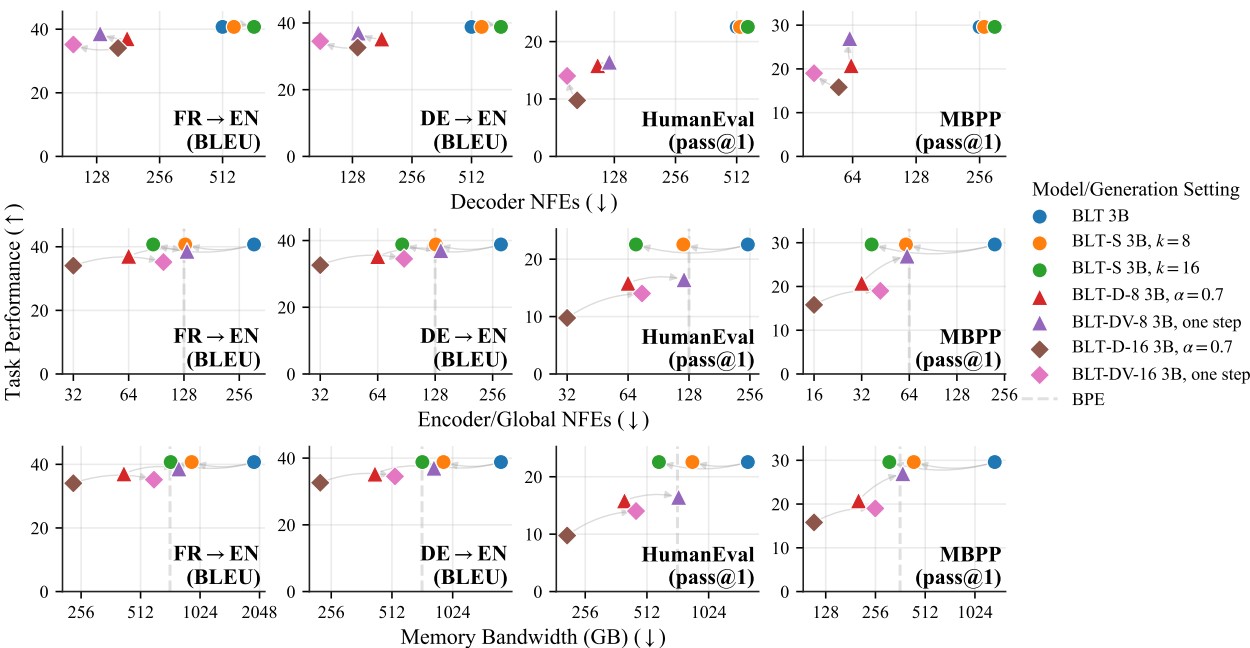

*Figure 4.* Generation task results for 3B-parameter variants of BLT, BLT-S, BLT-D, and BLT-DV. For space, we report results only for $k \in \{8, 16\}$ and $B \in \{8, 16\}$. Arrows indicate the same model evaluated with different inference methods. Verification (BLT-DV) enhances the task performance of BLT-D models, but increases global NFEs and memory bandwidth. Self-speculation (BLT-S) greatly improves BLT's speed, with no loss in task performance. BLT-D remains the fastest model/inference method overall.

immediately patching at each high-entropy byte, we propose a self-speculative decoding strategy, which we call **BLT-S (BLT Self-speculation)**. Here, the decoder always autoregressively generates up to a fixed window size $k$ regardless of entropy spikes, conditioning on the last available latent token. After producing a draft of $k$ bytes, the patcher segments the sequence and computes a full forward pass through $\mathcal{E}$, $\mathcal{G}$, and $\mathcal{D}$ to obtain new predictions. The model compares the drafted text to these predictions: if all bytes match, the draft is committed; if not, only the bytes up to the first mismatch are accepted. This iterative process advances by at least one verified byte per step and continues until the target sequence length is reached.

In our setup, verification requires an exact byte-wise match between drafted bytes and the model-verified bytes, and we only evaluate with greedy decoding. This procedure is inspired by speculative decoding but differs in that we validate the bytes themselves rather than their probability distributions; this makes our acceptance criteria stricter than standard speculative decoding. However, our setup is fully compatible with rejection sampling with different temperatures, but we leave explorations of these settings to future work. See Algorithm 2 for a detailed verification procedure.

Our method fundamentally differs from previous speculative decoding techniques, which typically employ a separate small model or additional layers for self-

verification (Leviathan et al., 2023; Zhang et al., 2024; Cai et al., 2024). In contrast, BLT-S leverages BLT's existing lightweight decoder ($\mathcal{D}$) for drafting, without introducing auxiliary models or new architectural overhead. By allowing the decoder to generate longer speculative windows, BLT-S increases the number of decoder NFEs, but reduces encoder and global model NFEs, leading to improved inference efficiency overall.

## 5.2. BLT Diffusion+Verification

Recall that the total training loss in Equation (7) includes $\mathcal{L}_{\text{clean}}$, the standard autoregressive loss. Since BLT-D is trained with a next-byte prediction objective, it can be run autoregressively using the same causal decoder masks as BLT. At inference time, the only adjustment needed is to apply the same decoder self-attention and cross-attention masks used in BLT. This design enables a new generation paradigm for BLT-D, where diffusion acts as the drafting mechanism, while autoregressive next-byte prediction serves as a verification step. We refer to the inference procedure that employs diffusion and verification as **BLT-DV (BLT Diffusion+Verification)**. After generating a block of bytes via diffusion, BLT-DV performs a full forward pass through $\mathcal{E}$, $\mathcal{G}$, and $\mathcal{D}$ with a causal mask to produce next-byte predictions. The model then verifies the block diffusion draft with the next-byte predictions using the same

procedure as in Algorithm 2.

Importantly, the same model parameters are used for both drafting and verification. The choice of block size $B$ and unmasking strategy determines the balance between generation speed and verification acceptance rate. Empirically, we found that combining one-step diffusion with verification yields the fastest inference. While one-step diffusion alone typically leads to rapid degradation in generation quality, the verification step effectively prevents this issue.

### 5.3. Evaluating Extensions on Generation Tasks

In Figure 4, we compare 3B-parameter BLT and BLT-D models, along with their respective versions incorporating our new inference extensions. This analysis examines decoder NFEs, encoder/global NFEs, memory bandwidth, and task performance on the same generation tasks described in Section 4.2. Section F and Section G also include additional inference hyperparameter sweeps and generation settings for all 1B and 3B model variants of BLT-S and BLT-DV, along with their verification acceptance rates. Overall, all BLT-D and BLT-DV models outperform BLT and BLT-S in terms of decoder NFEs. Notably, BLT-DV achieves slightly higher task performance than BLT-D without verification; however, this comes at the cost of increased encoder/global NFEs and thus memory bandwidth due to additional verification calls. BLT-S increases decoder NFEs when compared to BLT, but notably reduces encoder/global NFEs, resulting in improved efficiency and very competitive task performance. Despite these gains, BLT-D-8 and BLT-D-16 (without verification) remain the fastest models, though their task performance is somewhat diminished.

### 5.4. Likelihood-based Evaluations

In addition to generation tasks, we further evaluate BLT-D's verification ability on likelihood-based tasks. Since our diffusion models are also trained with a next-byte prediction objective, they inherently possess the ability to compute likelihoods for sequences. By applying a causal mask to the decoder, we can directly obtain these likelihood estimates. Importantly, this serves as a direct proxy for the quality of BLT-DV's verification mechanism, which uses the same masking patterns. We benchmark the performance of BLT and BLT-D models across five standard datasets: ARC-Easy (Clark et al., 2018), ARC-Challenge (Clark et al., 2018), PIQA (Bisk et al., 2019), HellaSwag (Zellers et al., 2019), and MMLU (Hendrycks et al., 2021) (see Table 1). The results show that BLT-D variants achieve scores approaching those of the BLT baseline, despite the added complexity of balancing next-byte prediction with the diffusion objective. This demonstrates that BLT-D's autoregressive capabilities remain robust and that the integration of block diffusion does not compromise autoregressive

*Table 1.* Performance comparison of 3B-parameter BLT and BLT-D models (block sizes: 4, 8, 16) across five benchmarks. While BLT-D variants exhibit a performance hit due to balancing next-byte prediction with the diffusion objective, the diffusion mechanism enables much faster inference for BLT-D.

| Benchmark | BLT 3B | BLT-D-4 3B | BLT-D-8 3B | BLT-D-16 3B |
|---|---|---|---|---|
| **ARC-Easy** | 74.33 | 72.39 | 70.95 | 66.89 |
| **ARC-Challenge** | 45.75 | 41.46 | 41.03 | 40.43 |
| **PIQA** | 79.38 | 79.60 | 78.02 | 76.93 |
| **HellaSwag** | 74.98 | 71.86 | 70.56 | 69.12 |
| **MMLU** | 41.15 | 39.07 | 38.29 | 37.08 |

performance on established language understanding and reasoning tasks. Overall, these findings suggest that BLT-D models can effectively combine block diffusion and next-byte prediction objectives, maintaining strong performance while ensuring high-quality generations.

## 6. Conclusion

In this paper, we introduced **BLT Diffusion (BLT-D)**, a byte-level language model that combines BLT's hierarchical latent tokenization with a block-wise diffusion objective to accelerate generation. BLT-D's new semi-autoregressive decoder design enables multiple future bytes to be generated in parallel, all while preserving BLT's dynamic patching and latent token representations. We also proposed two speculative-decoding–inspired extensions: **BLT Self-speculation (BLT-S)**, which uses BLT's own decoder to draft beyond normal patch boundaries before verification, and **BLT Diffusion+Verification (BLT-DV)**, which verifies diffusion drafts using autoregressive next-byte prediction. Each of these methods substantially reduces total model calls, narrowing the inference-efficiency gap between byte-level and subword-level models.

**Limitations and Future Work.** Here, we note our limitations and point out exciting avenues for future work. The main limitation of our evaluation is that we use network function evaluations (NFEs) and estimated memory bandwidth as proxy metrics for inference efficiency. NFEs are commonly reported in the discrete diffusion literature (see Lou et al. 2024; Arriola et al. 2025) because they isolate algorithmic efficiency from implementation-specific factors such as kernels, hardware utilization, batching strategy, and KV-cache management. Benchmarking BLT, BLT-D, BLT-S, and BLT-DV in a highly optimized inference implementation is therefore an important direction for future work. Other promising directions include experimenting with different patch sizes, tuning the balance between BLT-D's diffusion and next-byte prediction objectives, scaling training further—which may especially benefit diffusion language models (Ni et al., 2025)—and studying how decoder parameter allocation affects each BLT variant.

## Impact Statement

This paper presents work whose goal is to advance the field of machine learning. There are many potential societal consequences of our work, none of which we feel must be specifically highlighted here.

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

# A. Inference and Verification Algorithms

Algorithm 1 details BLT-D's generation procedure with text diffusion. Verification is optional and may be implemented as detailed in Algorithm 2. The `do_verify` branch is used for BLT-DV, introduced in Section 5; for BLT-D, `do_verify = False`. In our setup, verification requires an exact byte-wise match between drafted bytes and the model-verified bytes, and we only evaluate with greedy decoding. This procedure is inspired by speculative decoding but differs in that we validate the bytes themselves rather than their probability distributions; this makes our acceptance criteria stricter than standard speculative decoding. However, our setup is fully compatible with rejection sampling with different temperatures, but we leave explorations of these settings to future work.

---

**Algorithm 1** BLTDGeneration$(x, L, B, \text{do\_verify})$

---

**Input:** Initial byte sequence $x = [x_1; \ldots; x_N]$; generation length $L$; block size $B$; boolean `do_verify`
$l \leftarrow |x|$
**while** $l < N + L$ **do**
  **Patch Encoding:**
  Segment $x$ into $M$ patches via entropy-based patcher
  $\mathbf{T} \leftarrow \mathcal{E}(x); \mathbf{O} \leftarrow \mathcal{G}(\mathbf{T})$
  **Block Diffusion Decoding:**
  $x_{\text{block}} \leftarrow \{[\text{MASK}]\}^B$
  $x' \leftarrow [x_1; \ldots; x_l; x_{\text{block}}]$
  **while** $x'$ contains $[\text{MASK}]$ **do**
    $y \leftarrow \mathcal{D}(x'; \mathbf{O})$ {Bidirectional self-attention for block positions}
    Select $1 \leq k \leq B$ block positions to unmask {EB sampling or confidence-based}
    Replace selected $[\text{MASK}]$ positions in $x'$ with predictions from $y$
  **end while**
  **if** `do_verify` **then**
    $x \leftarrow \text{Verify}(x, x', l, B)$
  **else**
    $x \leftarrow x'$
  **end if**
  $l \leftarrow |x|$
**end while**
**Output:** Generated sequence $x$ of length $\geq N + L$

---

---

**Algorithm 2** Verify$(x, x', l, r)$

---

**Input:** current sequence $x$; candidate sequence $x'$; start index $l$; draft length $r$
Segment $x'$ into $M'$ patches via entropy-based patcher
$\mathbf{T}' \leftarrow \mathcal{E}(x'); \mathbf{O}' \leftarrow \mathcal{G}(\mathbf{T}'); y \leftarrow \mathcal{D}(x'; \mathbf{O}')$ {$y_j$ denotes the greedy next-byte prediction after position $j$}
$i \leftarrow l + 1$
**while** $i \leq l + r$ **do**
  **if** $x_i' \neq y_{i-1}$ **then**
    $x_i \leftarrow y_{i-1}$ {Reject drafted byte; replace first mismatch}
    **break**
  **else**
    $x_i \leftarrow x_i'$ {Accept drafted byte}
  **end if**
  $i \leftarrow i + 1$
**end while**
**if** $i = l + r + 1$ **then**
  $x_i \leftarrow y_{i-1}$ {No mismatches; use free byte from next-byte prediction}
**end if**
**Output:** updated sequence $x$

---

# B. BLT-D Inference Attention Masks

In Figure 5, we provide a visualization of the decoder's cross-attention and self-attention masks during generation/inference. This example and visualization are consistent with the formal description of the masks provided in Section 3.1.

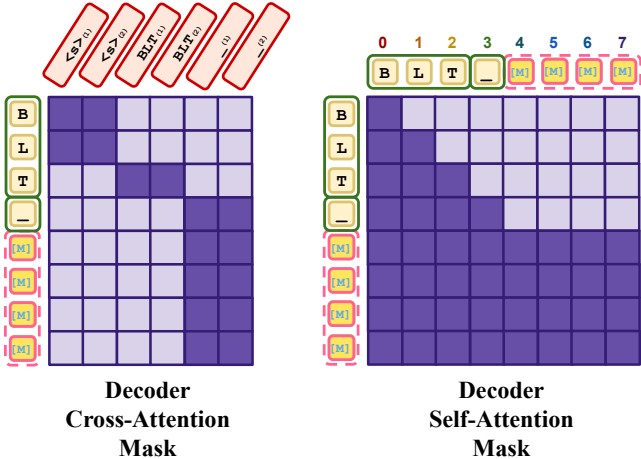

**Decoder Cross-Attention Mask**

**Decoder Self-Attention Mask**

*Figure 5.* **BLT-D attention masks during generation with block diffusion.** Before cross-attention, latent tokens are first split into multiple representations via a linear transformation and splitting function (described in detail in Pagnoni et al. 2025). Within the cross-attention, each byte attends to the representations of the previous latent token, except for the last byte of a patch, which may attend to its own latent token. In the self-attention, the clean prefix uses causal attention, and the corrupted/masked portion of the sequence uses bidirectional attention.

# C. Text Diffusion Unmasking Strategies

The choice of which bytes to unmask at each decoder forward pass affects both the generation quality and the degree of parallelism. We consider two unmasking strategies that differ in how they select masked positions for decoding.

**Confidence-based Unmasking.** The first strategy is confidence-based unmasking (Ghazvininejad et al., 2019). At each decoder step, the model predicts a distribution over the byte vocabulary for each masked position, and we measure confidence using the maximum predicted probability. All masked positions whose confidence exceeds a threshold $\alpha$ are decoded in parallel, while lower-confidence positions remain masked for subsequent steps. This approach prioritizes high-certainty predictions. If no position satisfies the threshold, the highest-confidence position is unmasked to ensure progress.

**Entropy-bounded Sampling.** The second strategy is entropy-bounded (EB) sampling (Ben-Hamu et al., 2025; Gat et al., 2025). At each decoder step, we compute the entropy of the predicted distribution for each masked token and sort masked positions in ascending order of entropy. Since mutual information among masked tokens is intractable to compute directly, we use an upper bound based on marginal entropies and select the largest subset of positions whose cumulative entropy does not exceed a threshold $\gamma$. The selected tokens are decoded in parallel, while the remaining tokens remain masked. This unmasking strategy may be combined with top-$p$ sampling to obtain diverse generations from the model. Like confidence-based unmasking, if no position satisfies the threshold, the lowest-entropy position is unmasked to ensure progress.

# D. BLT-D Training Data Preprocessing

In Figure 6, we provide a visualization of BLT-D's training data preprocessing, which includes both the block construction and diffusion corruption steps. This visualization is consistent with the description of the training data preprocessing pipeline outlined in Section 3.2.

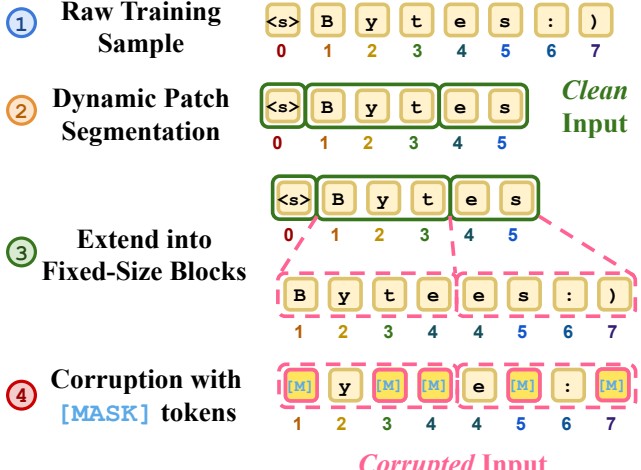

*Figure 6.* **BLT-D training data preprocessing.** (1) A raw training sample is loaded. (2) The entropy patcher segments the input dynamically; for illustration, only the first three patches are shown. This is referred to as the *clean* input. (3) All patches except the first are expanded into fixed-size blocks, containing bytes from future patches, with the original positional indices preserved. Allowing predictions beyond the patch enables BLT-D to draft beyond its average patch size during inference. (4) The blocks are corrupted with [MASK]s, resulting in the *corrupted* input.

# E. Architecture and Optimization Details

## E.1. Architecture Implementation Details

For all the BLT and BLT-D models we train, we maintain the same Transformer implementation details from the original BLT: the feed-forward layers use the SwiGLU activation function (Shazeer, 2020), all self-attention modules use rotary positional embeddings (RoPE, Su et al. 2023) with $\theta = 500000$ (Xiong et al., 2024), and layer normalization is done with RMSNorm (Zhang & Sennrich, 2019).

For self-attention in the encoder and global model, where the mask is fixed and follows a standard causal pattern with a fixed window, we use FlashAttention (Dao et al., 2022) with a window size of 512. For all cross-attention modules and the decoder's self-attention module, which requires carefully constructed custom masks that depend on the patch structure and vary per example, we use FlexAttention (Dong et al., 2025). FlexAttention streamlines the implementation of attention mechanisms with structured sparsity in PyTorch and allows users to define custom attention masks, all while achieving performance levels on par with specialized, manually optimized attention kernels.

## E.2. Pre-training Optimization and Hyperparameter Settings

All BLT/BLT-D 1B models are trained for 240,000 steps with a batch size of $2^{19}$ tokens per step (approximately 2 million bytes), and our 3B models are trained for 480,000 steps with a batch size of $2^{20}$ tokens per step (approximately 4 million bytes). All models use the AdamW optimizer (Loshchilov & Hutter, 2019) with $\beta_1 = 0.9$, $\beta_2 = 0.95$, and $\epsilon = 10^{-8}$. All models use a cosine learning rate schedule that linearly warms up to a peak learning rate of $4 \times 10^{-4}$ and decays to 0. The 1B models warm up to 2000 steps; the 3B models warm up to 4000 steps. We apply a weight decay of 0.1, and global gradient clipping at a threshold of 1.0.

# F. All 1B Model Results

In this section, we report results for all 1B models. Figure 7 and Figure 8 present the 1B counterparts of the generation-task results from Section 4.3 and Section 5.3 for BLT, BLT-D, BLT-S, and BLT-DV. Table 2 reports the likelihood-based evaluation results for the 1B models.

We also run a larger sweep over inference hyperparameters for the 1B models on the generation tasks. For BLT-D, we evaluate confidence-based unmasking with thresholds $\alpha \in \{0.5, 0.7\}$, as well as EB sampling with thresholds $\gamma \in \{0.8, 1.0\}$. For BLT-DV, we use more permissive settings that unmask more bytes per step; i.e., we *decrease* $\alpha$ for confidence-based unmasking or *increase* $\gamma$ for EB sampling. Specifically, we test $\alpha = 0.3$, $\gamma \in \{1.5, 2.0\}$, and one-step diffusion that unmasks all byte positions at once. For BLT-S, we use speculation windows $k \in \{4, 8, 16\}$. For BLT-S and BLT-DV, we also report the verification acceptance rate, defined as the fraction of drafted bytes that are accepted after verification. Table 3, Table 4, Table 5, and Table 6 report results on French-to-English translation, German-to-English translation, HumanEval, and MBPP, respectively.

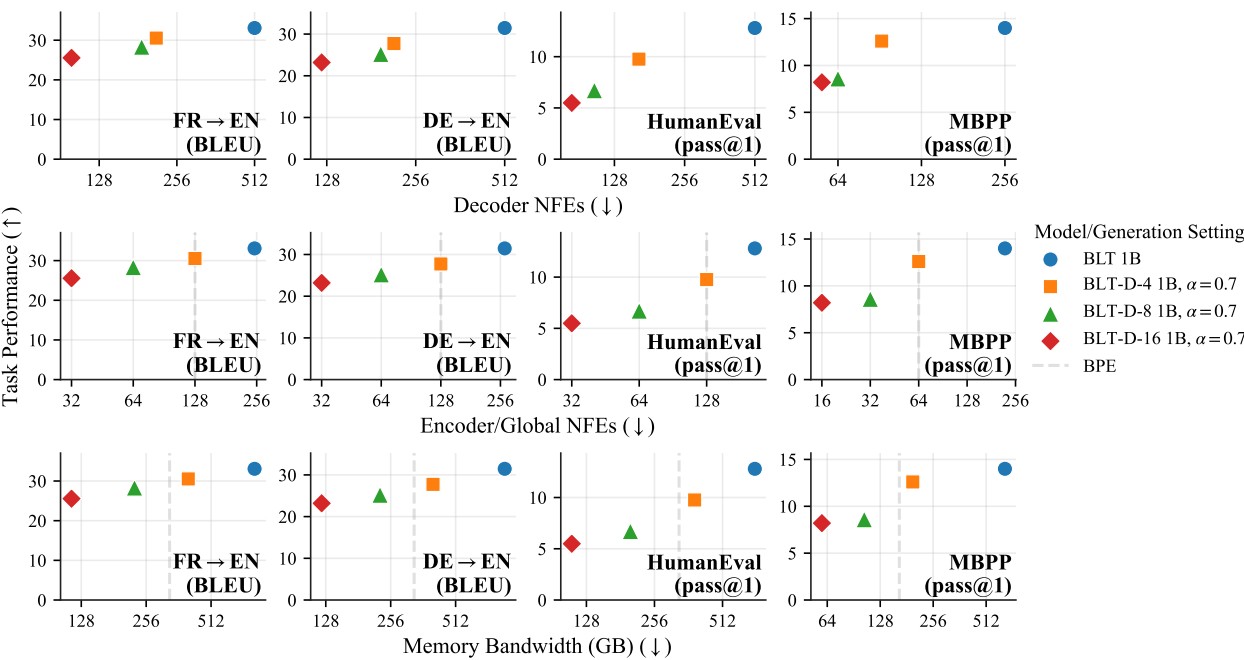

*Figure 7.* Generation task results of 1B-parameter variants of BLT, BLT-D-4, BLT-D-8, and BLT-D-16. Higher is better for task performance; lower is better for NFEs and memory bandwidth. The NFEs and memory bandwidth for a BPE model matching BLT's global model size are shown as a dashed line.

*Table 2.* Performance comparison of BLT and BLT-D (block sizes 4, 8, 16) at 1B parameters across five benchmarks.

| Benchmark | BLT 1B | BLT-D-4 1B | BLT-D-8 1B | BLT-D-16 1B |
|---|---|---|---|---|
| **ARC-Easy** | 63.21 | 60.76 | 61.06 | 59.83 |
| **ARC-Challenge** | 34.94 | 32.96 | 34.16 | 32.88 |
| **PIQA** | 75.46 | 74.48 | 73.56 | 72.36 |
| **HellaSwag** | 60.17 | 59.06 | 58.34 | 57.13 |
| **MMLU** | 33.90 | 33.60 | 32.28 | 32.09 |

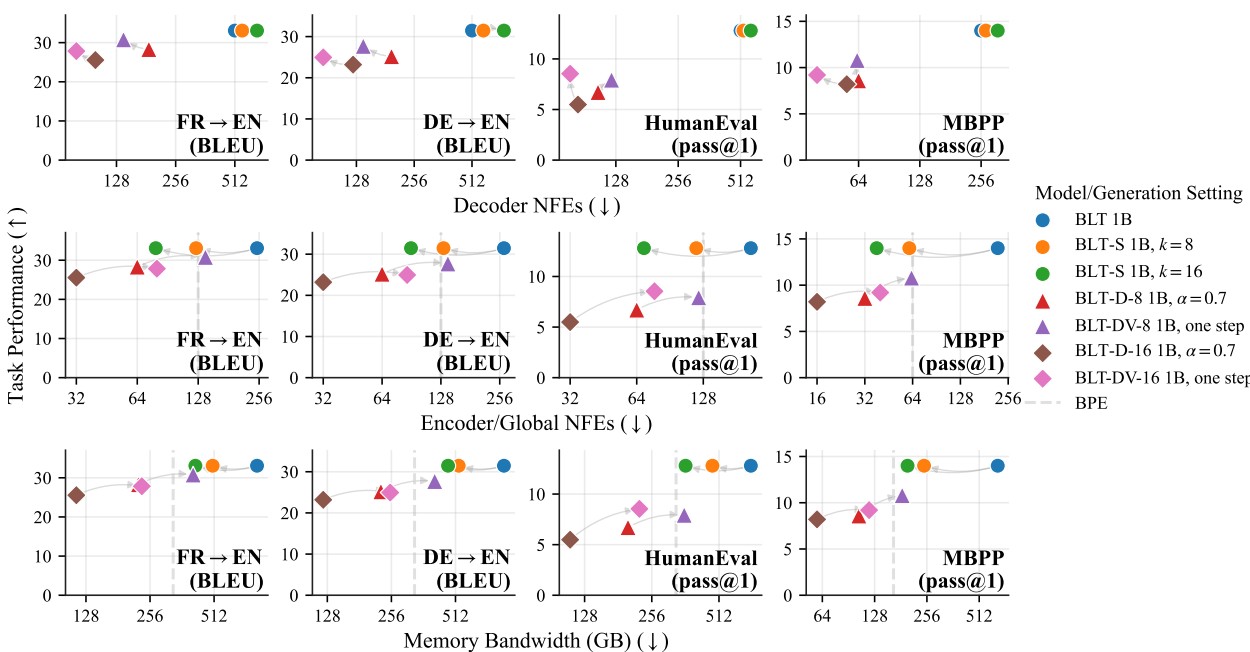

*Figure 8.* Generation task results for 1B-parameter variants of BLT, BLT-S, BLT-D, and BLT-DV. For space, we report results only for $k \in \{8, 16\}$ and $B \in \{8, 16\}$. Arrows indicate the same model evaluated with different inference methods.

*Table 3.* Full **French-to-English** translation results for **1B-parameter** models across various generation settings.

| Model | Generation Setting | BLEU | Diffusion/Speculation Sampling Strategy | Acceptance Rate (%) | Decoder NFEs | Global NFEs | Memory Bandwidth (GB) | Memory Decrease vs. BLT (%) |
|---|---|---|---|---|---|---|---|---|
| BLT 1B | BLT (AR) | 33.08 | — | — | 512 | 250 | 814.95 | — |
| | BLT-S (AR+self-speculation) | 33.08 | $k = 4$ | 96.77 | 526 | 212 | 719.45 | 11.72 |
| | | | $k = 8$ | 91.14 | 558 | 125 | 504.08 | 38.15 |
| | | | $k = 16$ | 76.93 | 664 | 79 | 418.26 | 48.68 |
| BLT-D-4 1B | BLT-D (diffusion only) | 30.01 | Confidence-based, $\alpha = 0.5$ | — | 184 | 128 | 392.17 | 51.88 |
| | | 30.53 | Confidence-based, $\alpha = 0.7$ | — | 213 | 128 | 401.38 | 50.75 |
| | | 30.59 | EB sampling, $\gamma = 0.8$ | — | 261 | 128 | 416.32 | 48.91 |
| | | 30.68 | EB sampling, $\gamma = 1.0$ | — | 249 | 128 | 412.68 | 49.36 |
| | BLT-DV (diffusion+verification) | 32.65 | Confidence-based, $\alpha = 0.3$ | 93.40 | 239 | 217 | 642.00 | 21.22 |
| | | | EB sampling, $\gamma = 1.5$ | 94.41 | 299 | 215 | 656.19 | 19.48 |
| | | | EB sampling, $\gamma = 2.0$ | 94.38 | 284 | 215 | 651.69 | 20.03 |
| | | | one step | 92.55 | 218 | 218 | 639.70 | 21.50 |
| BLT-D-8 1B | BLT-D (diffusion only) | 26.70 | Confidence-based, $\alpha = 0.5$ | — | 147 | 64 | 213.50 | 73.80 |
| | | 28.32 | Confidence-based, $\alpha = 0.7$ | — | 187 | 64 | 226.00 | 72.27 |
| | | 28.11 | EB sampling, $\gamma = 0.8$ | — | 259 | 64 | 248.91 | 69.46 |
| | | 28.14 | EB sampling, $\gamma = 1.0$ | — | 244 | 64 | 243.95 | 70.07 |
| | BLT-DV (diffusion+verification) | 30.80 | Confidence-based, $\alpha = 0.3$ | 83.81 | 176 | 134 | 406.37 | 50.14 |
| | | | EB sampling, $\gamma = 1.5$ | 86.91 | 272 | 130 | 425.34 | 47.81 |
| | | | EB sampling, $\gamma = 2.0$ | 86.69 | 253 | 130 | 420.33 | 48.42 |
| | | | one step | 80.34 | 139 | 139 | 408.23 | 49.91 |
| BLT-D-16 1B | BLT-D (diffusion only) | 23.68 | Confidence-based, $\alpha = 0.5$ | — | 77 | 32 | 107.92 | 86.76 |
| | | 25.55 | Confidence-based, $\alpha = 0.7$ | — | 100 | 32 | 115.34 | 85.85 |
| | | 25.49 | EB sampling, $\gamma = 0.8$ | — | 179 | 32 | 140.24 | 82.79 |
| | | 25.44 | EB sampling, $\gamma = 1.0$ | — | 167 | 32 | 136.42 | 83.26 |
| | BLT-DV (diffusion+verification) | 27.84 | Confidence-based, $\alpha = 0.3$ | 82.49 | 112 | 74 | 230.57 | 71.71 |
| | | | EB sampling, $\gamma = 1.5$ | 86.53 | 201 | 71 | 248.80 | 69.47 |
| | | | EB sampling, $\gamma = 2.0$ | 86.27 | 184 | 71 | 244.13 | 70.04 |
| | | | one step | 77.19 | 80 | 80 | 234.33 | 71.25 |

*Table 4.* Full **German-to-English** translation results for **1B-parameter** models across various generation settings.

| Model | Generation Setting | BLEU | Diffusion/Speculation Sampling Strategy | Acceptance Rate (%) | Decoder NFEs | Global NFEs | Memory Bandwidth (GB) | Memory Decrease vs. BLT (%) |
|---|---|---|---|---|---|---|---|---|
| BLT 1B | BLT (AR) | 31.46 | — | — | 512 | 269 | 864.76 | — |
| | BLT-S (AR+self-speculation) | 31.46 | $k = 4$
$k = 8$
$k = 16$ | 95.08
86.09
67.51 | 534
587
751 | 215
132
90 | 729.53
530.01
472.52 | 15.64
38.71
45.36 |
| BLT-D-4 1B | BLT-D (diffusion only) | 27.30
27.73
27.85
27.99 | Confidence-based, $\alpha = 0.5$
Confidence-based, $\alpha = 0.7$
EB sampling, $\gamma = 0.8$
EB sampling, $\gamma = 1.0$ | —
—
—
— | 189
216
259
247 | 128
128
128
128 | 393.58
402.31
415.71
411.90 | 54.49
53.48
51.93
52.37 |
| | BLT-DV (diffusion+verification) | 29.56 | Confidence-based, $\alpha = 0.3$
EB sampling, $\gamma = 1.5$
EB sampling, $\gamma = 2.0$
one step | 93.45
94.42
94.33
93.36 | 238
284
273
217 | 217
215
215
217 | 641.53
651.55
648.30
635.20 | 25.81
24.66
25.03
26.55 |
| BLT-D-8 1B | BLT-D (diffusion only) | 24.20
25.20
24.77
24.92 | Confidence-based, $\alpha = 0.5$
Confidence-based, $\alpha = 0.7$
EB sampling, $\gamma = 0.8$
EB sampling, $\gamma = 1.0$ | —
—
—
— | 157
195
252
238 | 64
64
64
64 | 216.67
228.50
246.55
242.16 | 74.94
73.58
71.49
72.00 |
| | BLT-DV (diffusion+verification) | 27.71 | Confidence-based, $\alpha = 0.3$
EB sampling, $\gamma = 1.5$
EB sampling, $\gamma = 2.0$
one step | 82.94
85.35
85.16
80.41 | 188
278
260
139 | 135
132
132
139 | 414.05
433.24
428.32
408.96 | 52.12
49.90
50.47
52.71 |
| BLT-D-16 1B | BLT-D (diffusion only) | 21.87
23.19
22.66
22.66 | Confidence-based, $\alpha = 0.5$
Confidence-based, $\alpha = 0.7$
EB sampling, $\gamma = 0.8$
EB sampling, $\gamma = 1.0$ | —
—
—
— | 94
123
208
195 | 32
32
32
32 | 113.32
122.48
149.20
145.24 | 86.90
85.84
82.75
83.20 |
| | BLT-DV (diffusion+verification) | 24.96 | Confidence-based, $\alpha = 0.3$
EB sampling, $\gamma = 1.5$
EB sampling, $\gamma = 2.0$
one step | 76.51
80.15
79.93
71.75 | 144
245
226
86 | 80
76
77
86 | 256.03
278.02
272.54
252.74 | 70.39
67.85
68.48
70.77 |

*Table 5.* Full **HumanEval** task results for **1B-parameter** models across various generation settings.

| Model | Generation Setting | PASS@1 | Diffusion/Speculation Sampling Strategy | Acceptance Rate (%) | Decoder NFEs | Global NFEs | Memory Bandwidth (GB) | Memory Decrease vs. BLT (%) |
|---|---|---|---|---|---|---|---|---|
| BLT 1B | BLT (AR) | 12.80 | — | — | 512 | 210 | 711.20 | — |
| | BLT-S (AR+self-speculation) | 12.80 | $k = 4$
$k = 8$
$k = 16$ | 98.77
96.45
88.92 | 517
529
574 | 208
119
69 | 707.67
478.64
362.77 | 0.50
32.70
48.99 |
| BLT-D-4 1B | BLT-D (diffusion only) | 8.54
9.76
10.37
10.37 | Confidence-based, $\alpha = 0.5$
Confidence-based, $\alpha = 0.7$
EB sampling, $\gamma = 0.8$
EB sampling, $\gamma = 1.0$ | —
—
—
— | 147
163
195
185 | 128
128
128
128 | 380.26
385.31
395.62
392.50 | 46.53
45.82
44.37
44.81 |
| | BLT-DV (diffusion+verification) | 9.15 | Confidence-based, $\alpha = 0.3$
EB sampling, $\gamma = 1.5$
EB sampling, $\gamma = 2.0$
one step | 97.76
98.22
98.22
97.69 | 213
239
231
209 | 209
208
208
209 | 613.63
619.73
617.23
612.52 | 13.72
12.86
13.21
13.88 |
| BLT-D-8 1B | BLT-D (diffusion only) | 6.71
6.71
7.93
7.93 | Confidence-based, $\alpha = 0.5$
Confidence-based, $\alpha = 0.7$
EB sampling, $\gamma = 0.8$
EB sampling, $\gamma = 1.0$ | —
—
—
— | 87
105
155
144 | 64
64
64
64 | 194.69
200.35
215.99
212.55 | 72.62
71.83
69.63
70.11 |
| | BLT-DV (diffusion+verification) | 7.93 | Confidence-based, $\alpha = 0.3$
EB sampling, $\gamma = 1.5$
EB sampling, $\gamma = 2.0$
one step | 93.73
95.25
95.15
93.13 | 131
180
169
122 | 121
119
119
122 | 358.72
369.54
366.22
357.62 | 49.56
48.04
48.51
49.72 |
| BLT-D-16 1B | BLT-D (diffusion only) | 3.66
5.49
6.10
5.49 | Confidence-based, $\alpha = 0.5$
Confidence-based, $\alpha = 0.7$
EB sampling, $\gamma = 0.8$
EB sampling, $\gamma = 1.0$ | —
—
—
— | 61
84
148
137 | 32
32
32
32 | 102.82
110.27
130.41
126.72 | 85.54
84.50
81.66
82.18 |
| | BLT-DV (diffusion+verification) | 8.54 | Confidence-based, $\alpha = 0.3$
EB sampling, $\gamma = 1.5$
EB sampling, $\gamma = 2.0$
one step | 81.72
86.99
86.75
78.92 | 97
180
164
77 | 74
70
70
77 | 225.05
240.02
235.13
225.34 | 68.36
66.25
66.94
68.32 |

*Table 6.* Full **MBPP** task results for **1B-parameter** models across various generation settings.

| Model | Generation Setting | PASS@1 | Diffusion/Speculation Sampling Strategy | Acceptance Rate (%) | Decoder NFEs | Global NFEs | Memory Bandwidth (GB) | Memory Decrease vs. BLT (%) |
|---|---|---|---|---|---|---|---|---|
| BLT 1B | BLT (AR) | 14.00 | — | — | 256 | 220 | 654.61 | — |
| | BLT-S (AR+self-speculation) | 14.00 | $k = 4$ | 98.02 | 261 | 105 | 358.79 | 45.19 |
| | | | $k = 8$ | 94.54 | 270 | 61 | 246.50 | 62.34 |
| | | | $k = 16$ | 82.43 | 309 | 38 | 198.05 | 69.75 |
| BLT-D-4 1B | BLT-D (diffusion only) | 9.60 | Confidence-based, $\alpha = 0.5$ | — | 79 | 64 | 191.88 | 70.69 |
| | | 12.60 | Confidence-based, $\alpha = 0.7$ | — | 92 | 64 | 195.97 | 70.06 |
| | | 12.40 | EB sampling, $\gamma = 0.8$ | — | 110 | 64 | 201.84 | 69.17 |
| | | 12.00 | EB sampling, $\gamma = 1.0$ | — | 105 | 64 | 200.09 | 69.43 |
| | BLT-DV (diffusion+verification) | 13.80 | Confidence-based, $\alpha = 0.3$ | 96.42 | 109 | 106 | 312.00 | 52.34 |
| | | | EB sampling, $\gamma = 1.5$ | 97.30 | 132 | 105 | 317.04 | 51.57 |
| | | | EB sampling, $\gamma = 2.0$ | 97.10 | 126 | 105 | 315.65 | 51.78 |
| | | | one step | 96.18 | 106 | 106 | 311.33 | 52.44 |
| BLT-D-8 1B | BLT-D (diffusion only) | 6.40 | Confidence-based, $\alpha = 0.5$ | — | 50 | 32 | 99.46 | 84.81 |
| | | 8.60 | Confidence-based, $\alpha = 0.7$ | — | 64 | 32 | 103.89 | 84.13 |
| | | 7.60 | EB sampling, $\gamma = 0.8$ | — | 92 | 32 | 112.74 | 82.78 |
| | | 7.60 | EB sampling, $\gamma = 1.0$ | — | 86 | 32 | 110.91 | 83.06 |
| | BLT-DV (diffusion+verification) | 10.80 | Confidence-based, $\alpha = 0.3$ | 91.56 | 68 | 62 | 184.69 | 71.79 |
| | | | EB sampling, $\gamma = 1.5$ | 93.79 | 103 | 61 | 192.32 | 70.62 |
| | | | EB sampling, $\gamma = 2.0$ | 94.00 | 95 | 61 | 189.54 | 71.05 |
| | | | one step | 90.60 | 63 | 63 | 184.50 | 71.82 |
| BLT-D-16 1B | BLT-D (diffusion only) | 5.60 | Confidence-based, $\alpha = 0.5$ | — | 40 | 16 | 54.48 | 91.68 |
| | | 8.20 | Confidence-based, $\alpha = 0.7$ | — | 56 | 16 | 59.66 | 90.89 |
| | | 8.00 | EB sampling, $\gamma = 0.8$ | — | 89 | 16 | 69.91 | 89.32 |
| | | 8.20 | EB sampling, $\gamma = 1.0$ | — | 82 | 16 | 67.74 | 89.65 |
| | BLT-DV (diffusion+verification) | 9.20 | Confidence-based, $\alpha = 0.3$ | 79.32 | 53 | 38 | 117.97 | 81.98 |
| | | | EB sampling, $\gamma = 1.5$ | 86.31 | 100 | 35 | 125.17 | 80.88 |
| | | | EB sampling, $\gamma = 2.0$ | 85.85 | 91 | 36 | 122.99 | 81.21 |
| | | | one step | 75.34 | 40 | 40 | 119.00 | 81.82 |

# G. All 3B Model Results

In this section, we present the results of a larger sweep over inference hyperparameters for our 3B BLT-D, BLT-DV, and BLT-S models on the generation tasks. For BLT-D, we evaluate confidence-based unmasking with thresholds $\alpha \in \{0.5, 0.7\}$, as well as EB sampling with thresholds $\gamma \in \{0.8, 1.0\}$. For BLT-DV, we use more permissive settings that unmask more bytes per step; i.e., we *decrease* $\alpha$ for confidence-based unmasking or *increase* $\gamma$ for EB sampling. Specifically, we test $\alpha = 0.3$, $\gamma \in \{1.5, 2.0\}$, and one-step diffusion that unmasks all byte positions at once. For BLT-S, we use speculation windows $k \in \{4, 8, 16\}$. For BLT-S and BLT-DV, we also report the verification acceptance rate, defined as the fraction of drafted bytes that are accepted after verification. Table 7, Table 8, Table 9, and Table 10 report results on French-to-English translation, German-to-English translation, HumanEval, and MBPP, respectively.

We further evaluate our methods on document-level German-to-English translation using the WMT 2018 news test set (Bojar et al., 2018), consisting of 347 documents with an average target length of approximately 910 bytes. We use confidence-based unmasking with $\alpha = 0.7$ for BLT-D and one-step diffusion for BLT-DV. Results are reported in Table 11. Across this longer-output setting, BLT-D maintains large efficiency gains: BLT-D-8 reduces runtime by 58% while closely matching BLT in BLEU, and BLT-D-16 reduces runtime by 73% with a small decrease in translation quality. BLT-DV improves the quality–efficiency trade-off further, with both BLT-DV-8 and BLT-DV-16 outperforming BLT in BLEU while retaining substantial runtime speedups.

In this set of results, we also provide another estimate of end-to-end latency. We benchmarked the forward-pass latency of each model component (encoder, global Transformer, decoder) on H200 GPUs, using `torch.cuda.synchronize()` at all timing boundaries. We discarded the first 5 passes as warm-up and computed the mean over 10 subsequent passes. Total runtime was extrapolated by multiplying each component's mean per-pass latency by the corresponding number of NFEs. This result is reported as "estimated runtime" in Table 7, Table 8, Table 9, Table 10, and Table 11.

*Table 7.* Full **French-to-English** translation results for **3B-parameter** models across various generation settings.

| Model | Generation Setting | BLEU | Diffusion/Speculation Sampling Strategy | Acceptance Rate (%) | Decoder NFEs | Global NFEs | Memory Bandwidth (GB) | Memory Decrease vs. BLT (%) | Estimated Runtime (s) | Runtime Decrease vs. BLT (%) |
|---|---|---|---|---|---|---|---|---|---|---|
| BLT 3B | BLT (AR) | 40.72 | — | — | 512 | 308 | 1920.99 | — | 21.06 | — |
| | BLT-S (AR+self-speculation) | 40.72 | $k = 4$ | 94.93 | 534 | 215 | 1395.99 | 27.33 | 16.89 | 19.79 |
| | | | $k = 8$ | 87.16 | 580 | 130 | 928.73 | 51.65 | 13.41 | 36.35 |
| | | | $k = 16$ | 69.93 | 724 | 87 | 727.17 | 62.15 | 13.14 | 37.61 |
| BLT-D-4 3B | BLT-D (diffusion only) | 37.75 | Confidence-based, $\alpha = 0.5$ | — | 185 | 128 | 787.36 | 59.01 | 8.41 | 60.08 |
| | | 38.09 | Confidence-based, $\alpha = 0.7$ | — | 216 | 128 | 797.58 | 58.48 | 8.79 | 58.25 |
| | | 37.79 | EB sampling, $\gamma = 0.8$ | — | 261 | 128 | 811.75 | 57.74 | 9.35 | 55.60 |
| | | 37.83 | EB sampling, $\gamma = 1.0$ | — | 250 | 128 | 808.18 | 57.93 | 9.22 | 56.25 |
| | BLT-DV (diffusion+verification) | 38.89 | Confidence-based, $\alpha = 0.3$ | 94.37 | 236 | 215 | 1300.92 | 32.28 | 13.20 | 37.35 |
| | | | EB sampling, $\gamma = 1.5$ | 95.37 | 290 | 213 | 1308.01 | 31.91 | 13.77 | 34.62 |
| | | | EB sampling, $\gamma = 2.0$ | 95.32 | 277 | 213 | 1304.23 | 32.11 | 13.61 | 35.39 |
| | | | one step | 93.12 | 217 | 217 | 1307.60 | 31.93 | 13.06 | 38.01 |
| BLT-D-8 3B | BLT-D (diffusion only) | 35.94 | Confidence-based, $\alpha = 0.5$ | — | 143 | 64 | 409.85 | 78.66 | 4.83 | 77.06 |
| | | 37.09 | Confidence-based, $\alpha = 0.7$ | — | 179 | 64 | 421.51 | 78.06 | 5.28 | 74.94 |
| | | 36.54 | EB sampling, $\gamma = 0.8$ | — | 249 | 64 | 443.89 | 76.89 | 6.15 | 70.82 |
| | | 36.60 | EB sampling, $\gamma = 1.0$ | — | 235 | 64 | 439.56 | 77.12 | 5.97 | 71.64 |
| | BLT-DV (diffusion+verification) | 38.66 | Confidence-based, $\alpha = 0.3$ | 86.25 | 166 | 130 | 797.43 | 58.49 | 8.27 | 60.74 |
| | | | EB sampling, $\gamma = 1.5$ | 89.34 | 251 | 126 | 802.07 | 58.25 | 9.13 | 56.64 |
| | | | EB sampling, $\gamma = 2.0$ | 88.79 | 236 | 127 | 801.11 | 58.30 | 8.99 | 57.30 |
| | | | one step | 84.63 | 133 | 133 | 799.94 | 58.36 | 8.00 | 62.01 |
| BLT-D-16 3B | BLT-D (diffusion only) | 31.64 | Confidence-based, $\alpha = 0.5$ | — | 123 | 32 | 221.58 | 88.47 | 3.05 | 85.50 |
| | | 34.05 | Confidence-based, $\alpha = 0.7$ | — | 162 | 32 | 233.87 | 87.83 | 3.54 | 83.20 |
| | | 33.75 | EB sampling, $\gamma = 0.8$ | — | 242 | 32 | 259.55 | 86.49 | 4.53 | 78.49 |
| | | 33.69 | EB sampling, $\gamma = 1.0$ | — | 229 | 32 | 255.41 | 86.70 | 4.37 | 79.25 |
| | BLT-DV (diffusion+verification) | 35.23 | Confidence-based, $\alpha = 0.3$ | 67.22 | 179 | 89 | 568.61 | 70.40 | 6.47 | 69.27 |
| | | | EB sampling, $\gamma = 1.5$ | 75.09 | 293 | 80 | 553.65 | 71.18 | 7.46 | 64.60 |
| | | | EB sampling, $\gamma = 2.0$ | 74.77 | 271 | 81 | 548.64 | 71.44 | 7.23 | 65.67 |
| | | | one step | 60.33 | 99 | 99 | 598.66 | 68.84 | 5.96 | 71.72 |

*Table 8.* Full **German-to-English** translation results for **3B-parameter** models across various generation settings.

| Model | Generation Setting | BLEU | Diffusion/Speculation Sampling Strategy | Acceptance Rate (%) | Decoder NFEs | Global NFEs | Memory Bandwidth (GB) | Memory Decrease vs. BLT (%) | Estimated Runtime (s) | Runtime Decrease vs. BLT (%) |
|---|---|---|---|---|---|---|---|---|---|---|
| BLT 3B | BLT (AR) | 38.82 | — | — | 512 | 283 | 1776.54 | — | 19.87 | — |
| | BLT-S (AR+self-speculation) | 38.82 | $k = 4$ | 95.67 | 531 | 214 | 1387.43 | 21.90 | 16.81 | 15.40 |
| | | | $k = 8$ | 87.82 | 576 | 129 | 922.56 | 48.07 | 13.31 | 33.02 |
| | | | $k = 16$ | 70.23 | 721 | 86 | 724.53 | 59.22 | 13.06 | 34.29 |
| BLT-D-4 3B | BLT-D (diffusion only) | 35.74 | Confidence-based, $\alpha = 0.5$ | — | 186 | 128 | 787.95 | 55.65 | 8.42 | 57.62 |
| | | 36.29 | Confidence-based, $\alpha = 0.7$ | — | 214 | 128 | 796.70 | 55.15 | 8.77 | 55.87 |
| | | 36.48 | EB sampling, $\gamma = 0.8$ | — | 247 | 128 | 807.51 | 54.55 | 9.18 | 53.81 |
| | | 36.53 | EB sampling, $\gamma = 1.0$ | — | 237 | 128 | 804.01 | 54.74 | 9.05 | 54.43 |
| | BLT-DV (diffusion+verification) | 37.46 | Confidence-based, $\alpha = 0.3$ | 94.35 | 235 | 215 | 1300.96 | 26.77 | 13.18 | 33.65 |
| | | | EB sampling, $\gamma = 1.5$ | 95.08 | 279 | 214 | 1307.33 | 26.41 | 13.68 | 31.14 |
| | | | EB sampling, $\gamma = 2.0$ | 94.99 | 268 | 214 | 1304.90 | 26.55 | 13.55 | 31.83 |
| | | | one step | 94.34 | 215 | 215 | 1294.63 | 27.13 | 12.94 | 34.90 |
| BLT-D-8 3B | BLT-D (diffusion only) | 33.83 | Confidence-based, $\alpha = 0.5$ | — | 146 | 64 | 411.05 | 76.86 | 4.87 | 75.50 |
| | | 35.29 | Confidence-based, $\alpha = 0.7$ | — | 180 | 64 | 421.74 | 76.26 | 5.29 | 73.37 |
| | | 35.41 | EB sampling, $\gamma = 0.8$ | — | 230 | 64 | 437.95 | 75.35 | 5.91 | 70.25 |
| | | 35.46 | EB sampling, $\gamma = 1.0$ | — | 217 | 64 | 433.80 | 75.58 | 5.75 | 71.06 |
| | BLT-DV (diffusion+verification) | 37.11 | Confidence-based, $\alpha = 0.3$ | 84.56 | 183 | 133 | 817.54 | 53.98 | 8.62 | 56.60 |
| | | | EB sampling, $\gamma = 1.5$ | 86.52 | 264 | 130 | 827.75 | 53.41 | 9.48 | 52.26 |
| | | | EB sampling, $\gamma = 2.0$ | 86.37 | 249 | 130 | 824.42 | 53.59 | 9.30 | 53.20 |
| | | | one step | 81.62 | 137 | 137 | 826.79 | 53.46 | 8.24 | 58.51 |
| BLT-D-16 3B | BLT-D (diffusion only) | 30.30 | Confidence-based, $\alpha = 0.5$ | — | 109 | 32 | 217.12 | 87.78 | 2.88 | 85.50 |
| | | 32.62 | Confidence-based, $\alpha = 0.7$ | — | 136 | 32 | 225.81 | 87.29 | 3.22 | 83.81 |
| | | 32.56 | EB sampling, $\gamma = 0.8$ | — | 206 | 32 | 247.99 | 86.04 | 4.08 | 79.44 |
| | | 32.48 | EB sampling, $\gamma = 1.0$ | — | 191 | 32 | 243.35 | 86.30 | 3.90 | 80.38 |
| | BLT-DV (diffusion+verification) | 34.52 | Confidence-based, $\alpha = 0.3$ | 74.49 | 161 | 83 | 524.04 | 70.50 | 5.96 | 69.99 |
| | | | EB sampling, $\gamma = 1.5$ | 77.89 | 263 | 79 | 534.83 | 69.89 | 7.04 | 64.58 |
| | | | EB sampling, $\gamma = 2.0$ | 77.66 | 245 | 79 | 530.54 | 70.14 | 6.81 | 65.71 |
| | | | one step | 70.44 | 88 | 88 | 529.76 | 70.18 | 5.29 | 73.35 |

*Table 9.* Full **HumanEval** task results for **3B-parameter** models across various generation settings.

| Model | Generation Setting | PASS@1 | Diffusion/Speculation Sampling Strategy | Acceptance Rate (%) | Decoder NFEs | Global NFEs | Memory Bandwidth (GB) | Memory Decrease vs. BLT (%) | Estimated Runtime (s) | Runtime Decrease vs. BLT (%) |
|---|---|---|---|---|---|---|---|---|---|---|
| BLT 3B | BLT (AR) | 22.56 | — | — | 512 | 250 | 1590.45 | — | 18.29 | — |
| | BLT-S (AR+self-speculation) | 22.56 | $k = 4$ | 98.68 | 518 | 208 | 1353.39 | 14.91 | 16.36 | 10.56 |
| | | | $k = 8$ | 95.96 | 532 | 120 | 853.11 | 46.36 | 12.33 | 32.58 |
| | | | $k = 16$ | 88.01 | 581 | 70 | 585.81 | 63.17 | 10.55 | 42.31 |
| BLT-D-4 3B | BLT-D (diffusion only) | 17.07 | Confidence-based, $\alpha = 0.5$ | — | 144 | 128 | 774.41 | 51.31 | 7.90 | 56.82 |
| | | 18.90 | Confidence-based, $\alpha = 0.7$ | — | 159 | 128 | 779.20 | 51.01 | 8.09 | 55.80 |
| | | 18.90 | EB sampling, $\gamma = 0.8$ | — | 188 | 128 | 788.50 | 50.42 | 8.45 | 53.83 |
| | | 18.90 | EB sampling, $\gamma = 1.0$ | — | 180 | 128 | 785.82 | 50.59 | 8.35 | 54.37 |
| | BLT-DV (diffusion+verification) | 18.90 | Confidence-based, $\alpha = 0.3$ | 97.97 | 214 | 208 | 1257.30 | 20.95 | 12.59 | 31.18 |
| | | | EB sampling, $\gamma = 1.5$ | 98.29 | 239 | 208 | 1262.78 | 20.60 | 12.90 | 29.49 |
| | | | EB sampling, $\gamma = 2.0$ | 98.26 | 232 | 208 | 1260.65 | 20.74 | 12.81 | 29.96 |
| | | | one step | 97.74 | 209 | 209 | 1258.33 | 20.88 | 12.57 | 31.26 |
| BLT-D-8 3B | BLT-D (diffusion only) | 10.37 | Confidence-based, $\alpha = 0.5$ | — | 88 | 64 | 392.51 | 75.32 | 4.15 | 77.32 |
| | | 15.85 | Confidence-based, $\alpha = 0.7$ | — | 106 | 64 | 398.04 | 74.97 | 4.37 | 76.10 |
| | | 15.24 | EB sampling, $\gamma = 0.8$ | — | 152 | 64 | 412.75 | 74.05 | 4.94 | 72.98 |
| | | 15.24 | EB sampling, $\gamma = 1.0$ | — | 142 | 64 | 409.53 | 74.25 | 4.82 | 73.66 |
| | BLT-DV (diffusion+verification) | 16.46 | Confidence-based, $\alpha = 0.3$ | 94.51 | 130 | 120 | 728.09 | 54.22 | 7.34 | 59.85 |
| | | | EB sampling, $\gamma = 1.5$ | 96.04 | 176 | 118 | 733.00 | 53.91 | 7.82 | 57.25 |
| | | | EB sampling, $\gamma = 2.0$ | 95.91 | 165 | 119 | 730.33 | 54.08 | 7.73 | 57.74 |
| | | | one step | 93.63 | 121 | 121 | 731.02 | 54.04 | 7.28 | 60.20 |
| BLT-D-16 3B | BLT-D (diffusion only) | 8.54 | Confidence-based, $\alpha = 0.5$ | — | 62 | 32 | 201.98 | 87.30 | 2.30 | 87.44 |
| | | 9.76 | Confidence-based, $\alpha = 0.7$ | — | 84 | 32 | 208.94 | 86.86 | 2.57 | 85.95 |
| | | 11.59 | EB sampling, $\gamma = 0.8$ | — | 143 | 32 | 227.97 | 85.67 | 3.30 | 81.94 |
| | | 10.98 | EB sampling, $\gamma = 1.0$ | — | 133 | 32 | 224.65 | 85.88 | 3.18 | 82.62 |
| | BLT-DV (diffusion+verification) | 14.02 | Confidence-based, $\alpha = 0.3$ | 83.61 | 94 | 72 | 445.22 | 72.01 | 4.60 | 74.83 |
| | | | EB sampling, $\gamma = 1.5$ | 87.77 | 172 | 69 | 449.96 | 71.71 | 5.43 | 70.32 |
| | | | EB sampling, $\gamma = 2.0$ | 87.61 | 155 | 69 | 445.39 | 72.00 | 5.22 | 71.47 |
| | | | one step | 80.68 | 75 | 75 | 453.15 | 71.51 | 4.51 | 75.33 |

*Table 10.* Full **MBPP** task results for **3B-parameter** models across various generation settings.

| Model | Generation Setting | PASS@1 | Diffusion/Speculation Sampling Strategy | Acceptance Rate (%) | Decoder NFEs | Global NFEs | Memory Bandwidth (GB) | Memory Decrease vs. BLT (%) | Estimated Runtime (s) | Runtime Decrease vs. BLT (%) |
|---|---|---|---|---|---|---|---|---|---|---|
| BLT 3B | BLT (AR) | 29.60 | — | — | 256 | 222 | 1349.34 | — | 13.78 | — |
| | BLT-S (AR+self-speculation) | 29.60 | $k = 4$ | 98.21 | 260 | 105 | 685.35 | 49.21 | 8.24 | 40.19 |
| | | | $k = 8$ | 94.84 | 269 | 61 | 436.89 | 67.62 | 6.25 | 54.63 |
| | | | $k = 16$ | 84.41 | 302 | 37 | 310.63 | 76.98 | 5.52 | 59.97 |
| BLT-D-4 3B | BLT-D (diffusion only) | 24.60 | Confidence-based, $\alpha = 0.5$ | — | 76 | 64 | 388.71 | 71.19 | 4.00 | 70.97 |
| | | 26.00 | Confidence-based, $\alpha = 0.7$ | — | 89 | 64 | 392.59 | 70.90 | 4.16 | 69.80 |
| | | 25.80 | EB sampling, $\gamma = 0.8$ | — | 107 | 64 | 398.39 | 70.48 | 4.38 | 68.18 |
| | | 25.80 | EB sampling, $\gamma = 1.0$ | — | 101 | 64 | 396.72 | 70.60 | 4.31 | 68.72 |
| | BLT-DV (diffusion+verification) | 27.20 | EB sampling, $\gamma = 1.5$ | 97.94 | 129 | 105 | 639.19 | 52.63 | 6.62 | 51.99 |
| | | | EB sampling, $\gamma = 2.0$ | 97.86 | 123 | 105 | 637.77 | 52.74 | 6.54 | 52.53 |
| | | | one step | 96.98 | 105 | 105 | 635.95 | 52.87 | 6.32 | 54.15 |
| BLT-D-8 3B | BLT-D (diffusion only) | 18.40 | Confidence-based, $\alpha = 0.5$ | — | 49 | 32 | 197.75 | 85.34 | 2.14 | 84.50 |
| | | 20.80 | Confidence-based, $\alpha = 0.7$ | — | 63 | 32 | 202.22 | 85.01 | 2.31 | 83.23 |
| | | 23.20 | EB sampling, $\gamma = 0.8$ | — | 88 | 32 | 210.29 | 84.42 | 2.62 | 80.98 |
| | | 22.40 | EB sampling, $\gamma = 1.0$ | — | 82 | 32 | 208.37 | 84.56 | 2.55 | 81.52 |
| | BLT-DV (diffusion+verification) | 27.00 | Confidence-based, $\alpha = 0.3$ | 92.68 | 67 | 61 | 373.68 | 72.31 | 3.74 | 72.82 |
| | | | EB sampling, $\gamma = 1.5$ | 94.85 | 99 | 60 | 376.71 | 72.08 | 4.09 | 70.29 |
| | | | EB sampling, $\gamma = 2.0$ | 94.65 | 92 | 60 | 375.13 | 72.20 | 4.01 | 70.92 |
| | | | one step | 91.87 | 62 | 62 | 374.64 | 72.24 | 3.73 | 72.93 |
| BLT-D-16 3B | BLT-D (diffusion only) | 10.60 | Confidence-based, $\alpha = 0.5$ | — | 39 | 16 | 103.64 | 92.32 | 1.25 | 90.94 |
| | | 15.80 | Confidence-based, $\alpha = 0.7$ | — | 55 | 16 | 108.78 | 91.94 | 1.45 | 89.50 |
| | | 15.60 | EB sampling, $\gamma = 0.8$ | — | 88 | 16 | 119.48 | 91.15 | 1.86 | 86.53 |
| | | 15.60 | EB sampling, $\gamma = 1.0$ | — | 81 | 16 | 117.21 | 91.31 | 1.77 | 87.16 |
| | BLT-DV (diffusion+verification) | 19.00 | Confidence-based, $\alpha = 0.3$ | 74.77 | 56 | 41 | 251.34 | 81.37 | 2.65 | 80.75 |
| | | | EB sampling, $\gamma = 1.5$ | 81.05 | 107 | 37 | 250.43 | 81.44 | 3.09 | 77.54 |
| | | | EB sampling, $\gamma = 2.0$ | 80.38 | 97 | 38 | 249.14 | 81.54 | 3.02 | 78.09 |
| | | | one step | 71.79 | 42 | 42 | 255.40 | 81.07 | 2.53 | 81.66 |

*Table 11.* Additional **WMT German-to-English** generation results for **3B-parameter** models across various generation settings.

| Model | Generation Setting | BLEU | Diffusion/Sampling Strategy | Acceptance Rate (%) | Decoder NFEs | Global NFEs | Memory Bandwidth (GB) | Memory Decrease vs. BLT (%) | Estimated Runtime (s) | Runtime Decrease vs. BLT (%) |
|---|---|---|---|---|---|---|---|---|---|---|
| BLT 3B | BLT (AR) | 33.99 | — | — | 2000 | 394 | 2881.85 | — | 43.64 | — |
| BLT-D-4 3B | BLT-D (diffusion only) | 36.84 | Confidence-based, $\alpha = 0.7$ | — | 700 | 500 | 3068.38 | -6.47 | 32.57 | 25.38 |
| | BLT-DV (diffusion+verification) | 34.53 | one-step diffusion | 93.70 | 844 | 844 | 5070.31 | -75.94 | 50.77 | -16.33 |
| BLT-D-8 3B | BLT-D (diffusion only) | 33.73 | Confidence-based, $\alpha = 0.7$ | — | 527 | 250 | 1590.78 | 44.80 | 18.48 | 57.66 |
| | BLT-DV (diffusion+verification) | 35.51 | one-step diffusion | 87.90 | 500 | 500 | 3005.43 | -4.29 | 30.09 | 31.05 |
| BLT-D-16 3B | BLT-D (diffusion only) | 32.14 | Confidence-based, $\alpha = 0.7$ | — | 465 | 125 | 859.96 | 70.16 | 11.74 | 73.09 |
| | BLT-DV (diffusion+verification) | 34.71 | one-step diffusion | 69.30 | 336 | 336 | 2020.82 | 29.88 | 20.23 | 53.64 |

# H. BLT-D Generation Analysis

In this section, we analyze the diversity and efficiency of unconditional generations produced by BLT-D models using entropy-bounded sampling as the unmasking strategy. Because entropy-bounded sampling can be combined with top-$p$ sampling, this setup allows us to sample diverse outputs while varying the amount of parallelism during block diffusion decoding. This analysis focuses on the block generation ability of BLT-D without autoregressive next-byte verification. For each model and sampling configuration, we generate text unconditionally from the start-of-sequence token until reaching a maximum length of 1k bytes.

To quantify diversity, we compute the word-level type-token ratio (TTR) of the generated text after whitespace tokenization, and compare it against the number of decoder network function evaluations (NFEs) required under varying entropy-bounded sampling thresholds ($\gamma$) and top-$p$ values. TTR serves as a simple proxy for lexical diversity, with higher values indicating a greater variety of unique words relative to the total word count. The resulting diversity–efficiency trade-off is shown in Figure 9.

Our results show a clear trend: as the number of decoder calls increases, the type-token ratio also increases. This suggests that more decoder forward passes are associated with the generation of more diverse text. Conversely, when the model produces repetitive or highly predictable text, it requires fewer decoder calls, reflecting the lower uncertainty and entropy in those outputs. This relationship highlights a key advantage of block diffusion decoding: it provides a mechanism to explore the trade-off between generation diversity and computational efficiency.

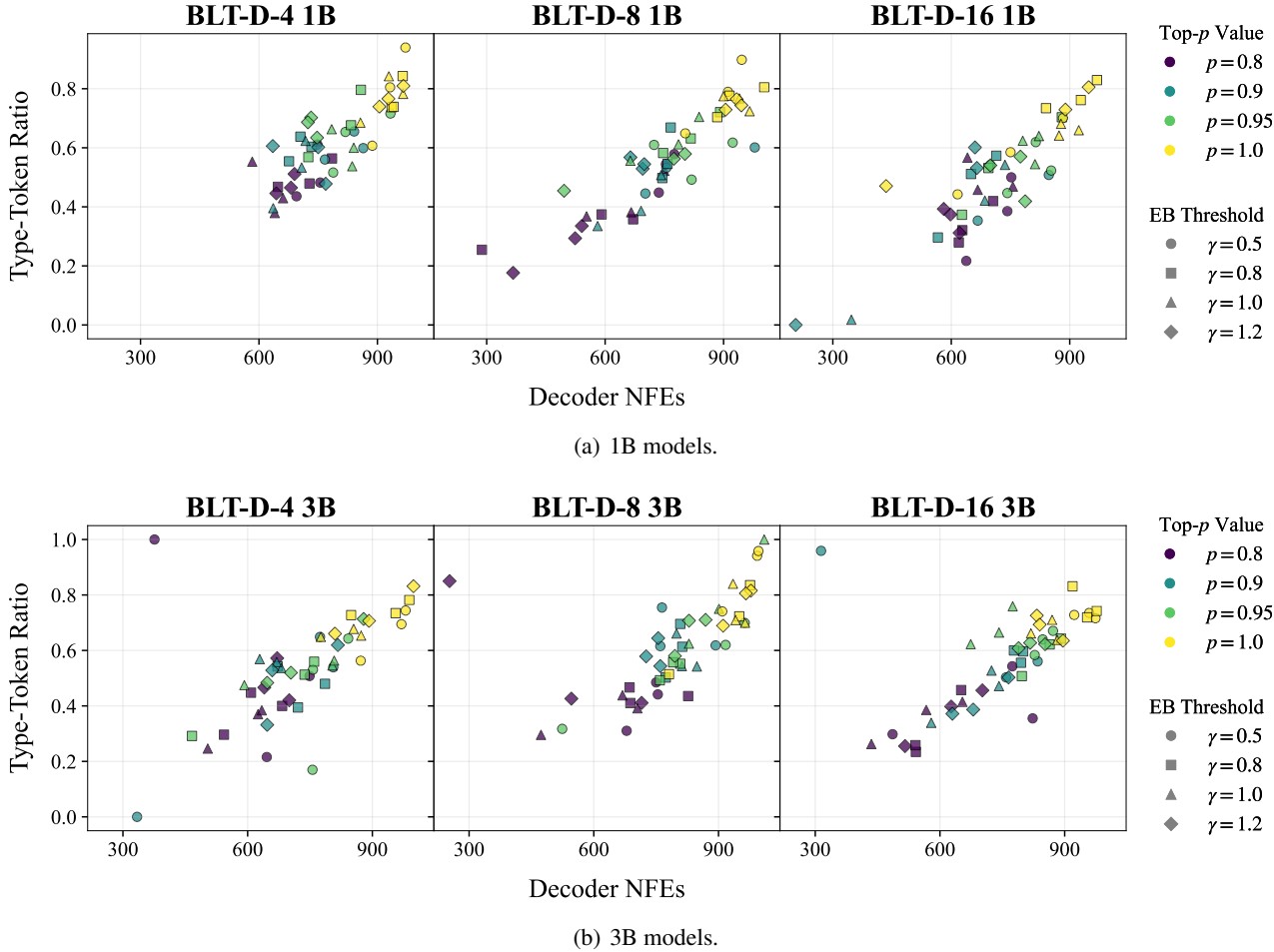

(a) 1B models.

(b) 3B models.

*Figure 9.* Type-token ratio increases with the number of decoder calls when generating text with BLT-D using entropy-bounded sampling with top-$p$ sampling. This indicates that more decoder passes yield greater diversity, while fewer passes correspond to more repetitive, predictable text. Block diffusion decoding enables exploration of this trade-off between generation diversity and computational efficiency.

# I. Scaling Analysis and Additional Evaluations

To better characterize the scaling behavior of BLT and the proposed BLT-D variants, we trained additional 400M-parameter models from scratch. Together with the 1B and 3B models, this yields three model scales for studying parameter scaling under a fixed training budget. All models in this analysis were trained on 1T tokens.

## I.1. Scaling Behavior on HellaSwag NLL

We fit single-variable power-law scaling curves to HellaSwag negative log-likelihood (NLL) as a function of model size. Specifically, for each architecture, we fit curves of the form

$$L(N) = A \cdot N^{-\alpha} + L_\infty, \tag{9}$$

where $N$ denotes the number of model parameters, $\alpha$ is the scaling exponent, $A$ is a multiplicative constant, and $L_\infty$ is the asymptotic loss floor. The resulting fits are summarized in Table 12, and the corresponding log-log curves are shown in Figure 10. Across all variants, the fitted curves achieve high coefficients of determination, with $R^2 \geq 0.97$. The estimated scaling exponents are also similar across architectures, ranging from approximately 0.020 to 0.023. This suggests that BLT and the BLT-D variants improve at comparable rates as parameter count increases under a fixed-token training regime.

At the same time, the fitted curves indicate a consistent offset in favor of the BLT baseline. BLT obtains the lowest extrapolated HellaSwag NLL at both projected scales, with predicted losses of 1.2306 at 7B parameters and 1.2135 at 13B parameters. BLT-D-4 and BLT-D-8 follow closely, while BLT-D-16 trails slightly. These results suggest that, although the architectures exhibit similar scaling exponents, BLT retains a persistent loss advantage within this parameter-scaling regime.

These extrapolations should be interpreted cautiously. The scaling curves are fit from only three model sizes and contain three free parameters, making the absolute extrapolated losses underconstrained. We therefore emphasize the relative trends and ordering across model variants rather than the precise numerical predictions at larger scales. More broadly, scaling behavior for diffusion language models remains less well characterized than for standard autoregressive language models. In particular, diffusion language models may be more sensitive to the data axis, and varying the number of training tokens could change the relative comparison between architectures. The results in this section should therefore be viewed as evidence about parameter scaling under a fixed 1T-token budget, rather than as a complete characterization of compute- or data-optimal scaling.

## I.2. Downstream Evaluations Across Scales

We also evaluate the models on translation tasks and likelihood-based benchmarks, with results shown in Figures 11 and 12. We omit coding benchmarks from this analysis because the 400M BLT baseline does not perform meaningfully on them, making comparisons at this scale uninformative.

Across both evaluation categories, all model families improve consistently with scale. This trend holds for BLT, BLT-D, and BLT-DV variants across the evaluated block sizes. On translation tasks, the gap between BLT and the BLT-D variants narrows as model size increases, and variants with larger diffusion block sizes appear to benefit more from additional parameters. These results indicate that the generative capabilities of the diffusion-based models improve substantially with scale.

The likelihood-based benchmarks show a less monotonic relative pattern: the gap between BLT and the diffusion-based variants narrows from 400M to 1B parameters, but widens again from 1B to 3B parameters. Nevertheless, the absolute performance of all model families improves with scale. We view the translation results as especially informative because they directly evaluate the generative behavior of the models, which is the primary mode of use for diffusion language models at inference time.

*Table 12.* Power-law scaling fits for HellaSwag NLL across 400M, 1B, and 3B model scales. Predicted losses are extrapolated to 7B and 13B parameters using the fitted curves.

| Model | $\alpha$ | $A$ | $L_\infty$ | $R^2$ | Pred. 7B | Pred. 13B |
|-------|-------|-------|-------|-------|-------|-------|
| BLT-D-4 | 0.0204 | 1.9686 | 0.0000 | 0.9780 | 1.2398 | 1.2243 |
| BLT-D-8 | 0.0222 | 2.0123 | 0.0235 | 0.9989 | 1.2405 | 1.2239 |
| BLT-D-16 | 0.0209 | 1.9676 | 0.0238 | 0.9986 | 1.2480 | 1.2323 |
| BLT | 0.0227 | 2.0594 | 0.0000 | 0.9689 | 1.2306 | 1.2135 |

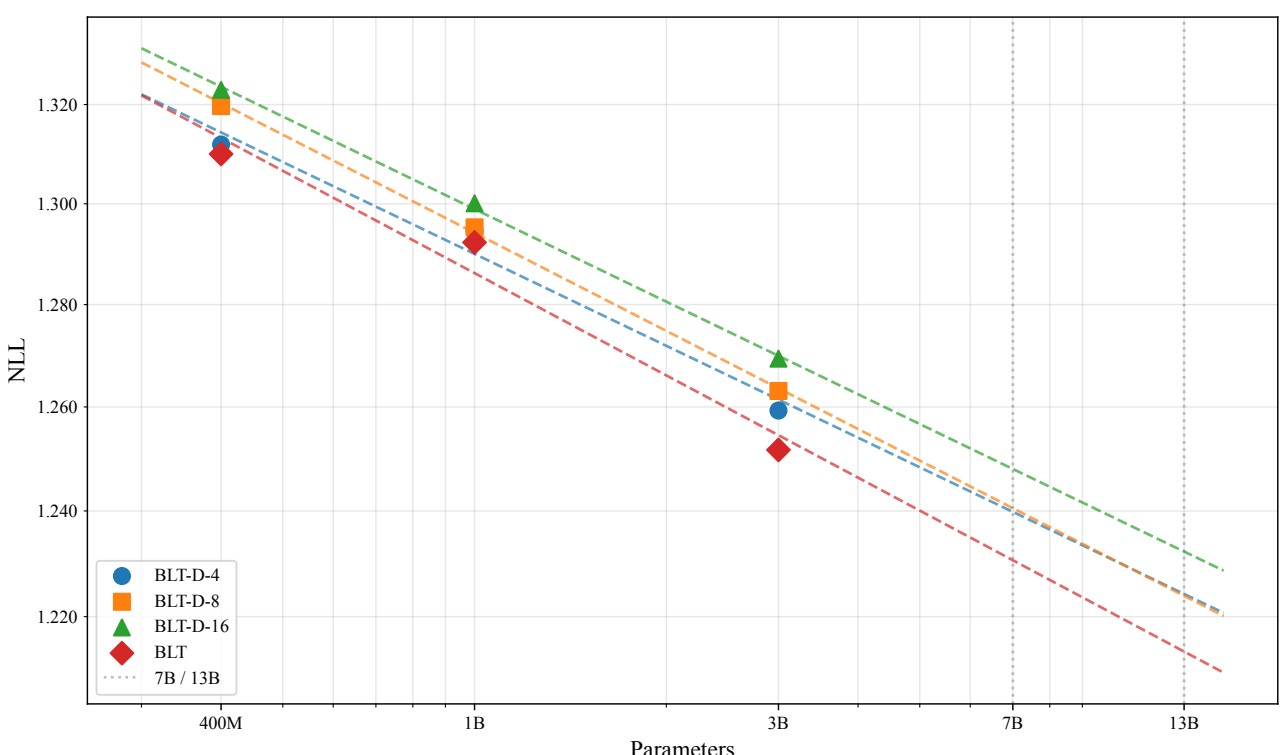

*Figure 10.* Log-log scaling curves for HellaSwag NLL as a function of model size. Each curve is fit using the 400M, 1B, and 3B checkpoints trained on 1T tokens.

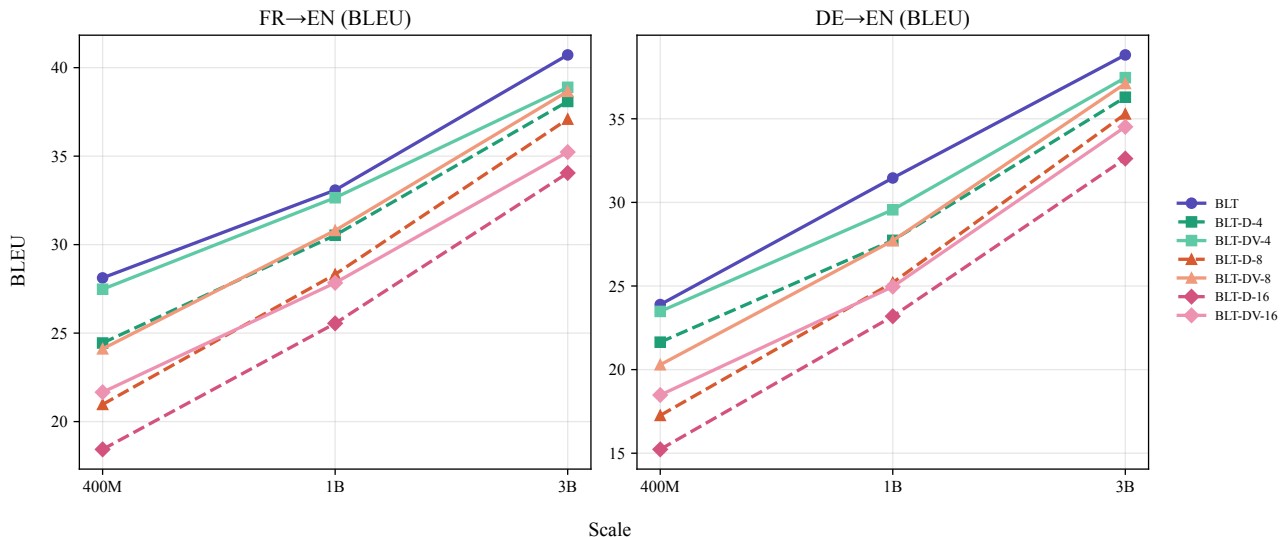

*Figure 11.* Translation performance across model scales for BLT, BLT-D, and BLT-DV variants. The performance gap between BLT and the diffusion-based variants narrows as model size increases.

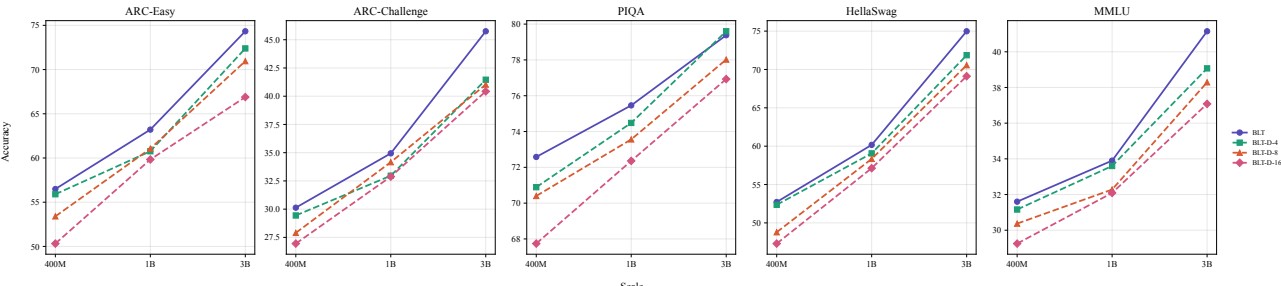

*Figure 12.* Likelihood-based benchmark results across model scales for BLT and BLT-D variants. All model families improve with scale, although the relative gaps between architectures vary across model sizes.

