# OpenReview forum: "Fast Byte Latent Transformer"
_ICML.cc/2026/Conference — ICML 2026 regular_

### Official Review · Reviewer_ErPH · 2026-02-17

**Soundness:** 4
**Presentation:** 4
**Significance:** 4
**Originality:** 3
**Overall Recommendation:** 4
**Confidence:** 5

**Summary:**

This paper proposes BLT Diffusion (BLT-D), a method that adopts diffusion generation to accelerate byte-level transformers (BLT). Compared with vanilla BLT, the encoder and the autoregressive backbone remain unchanged, while the local decoder is replaced from a semi-autoregressive head to a block-level diffusion head. Following the block-diffusion training procedure, BLT-D is able to generate bytes parallely to speedup language generation. Beyond BLT-D, this paper also introduces BLT-S and BLT-DV. BLT-S is a generation method that utilizes speculative decoding to accelerate BLT inference which follows the draft-then-verify pattern on bytes generation. BLT-DV takes the the advantage of BLT-D that the diffusion model can also generate bytes one-by-one in autoregression mode. The bytes are first drafted in diffusion mode, then verified in autoregression mode to achieve a better performance. This paper includes empirical results including the comparision of BLT, BLT-D, BLT-S, and BLT-DV at 3B scale on multiple tasks, e.g. translation and code generation. Experimental results show that BLT-D achieves notable speedups compared with the vanilla BLT, and BLT-S and BLT-DV can also accelerate the inference of byte-level transformers while maintaining most of the performance.

**Compliance With Llm Reviewing Policy:**

Affirmed.

**Final Justification:**

All weaknesses and questions are resolved during the discussion period. Since 4 is a positive score, I'm keeping my score unchanged.

**Key Questions For Authors:**

See weaknesses above.

**Limitations:**

yes

**Strengths And Weaknesses:**

## **Strength**
---

1. This paper solves a very important problem of byte-level transformers: the generation of bytes is generally slow and constrainted by the memory IO. By introducing the diffusion pattern to byte-level transformers, the generation speed is significantly higher, which unlocks the potential of real-world usage of BLTs. The change of model architecture is lightweight, as only the local decoder head is changed to a diffusion head, which makes the model easy to use and potentially orthogonal to future optimzations of BLTs.

2. This paper is well written in the main text with sufficient details included in the appendix. The paper presents the challenges of BLTs clearly, and the descriptions of the methods are easy to follow. The figures are drawn in a good manner while the visulizations of the results are clear. Training and generation details are included in the appendix, which makes the paper sound and solid.

3. This paper provides a large scale of contribution to the topic of accelerating BLTs. After proving that the BLT-D models can obtain higher inference speed with comparable task performance, this paper also build BLT-S and BLT-DV to further explore the potential of integrating speculative decoding to accelerate BLTs. The AR/Diffusion switchable feature of BLT-D models is compatible to self-speculative decoding, which provides another choice that a higher performance is in demand since speculative decoding is a lossless acceleration method w.r.t. BLT-D running in AR mode.

## **Weaknesses**
---

1. The "speedup" measured in this paper is the reduction of memory bandwidth instead of end-to-end wall-clock time speedups. Although LLM decoding is memory-bounded, the reduction of memory IO is only the theoretical acceleration upper-bound, and may not be achieved in most scenarios. Therefore, reporting the real-world end-to-end wall-clock time speedups is vital to show that the potential of BLT-D can actually lead to a faster inference.

2. This paper claims that BLT-D can achieve *some* degredation compared with BLT. However, there appears to be more than 5% performance loss on HumanEval in Figure 8. How large is the performance degredation intuitively? Since there is no other baselines included in the experiment section, the readers would be hard to judge whether such performance degredation is mild or severe. My suggestion is to add an BPE task performance baseline for a reference, which would make this more clear. I admit that training a 3B BPE model during the response period might not be possible, so providing more explanation on this topic is also acceptable. However, I highly recommend adding a BPE baseline if possible.

3. Following weakness 2, a scaling experiment with different training FLOPs is absent in this paper, and these results would provide more perspectives of BLT and BLT-D. As shown in the original BLT [1] paper, BLT only surpasses BPE under large training flops. The performance gap between BLT-D and BLT under various model sizes and training FLOPs remains unstudied. Do they also have similar scaling features with BLT? Note that this might not be a strict weakness (and you can overlook this suggestion if it is not possible during the response period), but adding these experiments can largely improve the experimental depth of this paper.

---
[1] Byte latent transformer: Patches scale better than tokens.

---

> ### Author Rebuttal · Authors · 2026-03-30
>
> We sincerely thank Reviewer ErPH for their positive and detailed review!
>
> ## Wall-clock time vs. memory bandwidth
> This is a great point. We have conducted real wall-clock time measurements and include them in our response to Reviewer pS5W. In short, the runtime speedups closely mirror the memory bandwidth reductions, confirming that our theoretical efficiency gains translate to practice. We will incorporate these results into the revised paper.
>
> ## Performance degradation and BPE baseline
> We appreciate this suggestion; a BPE reference point would indeed help readers contextualize the quality trade-off. While we cannot train a 3B BPE baseline during the rebuttal period, the closest available reference is from Pagnoni et al. at 8B scale, where BLT performs better than Llama 3 (e.g., DE→EN: 42.0 vs. 41.3; HumanEval: 35.4 vs. 31.1; MBPP: 41.8 vs. 40.2). **BLT's advantages over BPE tend to grow with scale,** so at our 3B scale the margin is likely smaller. We would likely see more performance gains for all of our model variants at a larger scale. We will add a more explicit discussion of this in the revised paper.
>
> Regarding why code tasks suffer more than translation: see our response to Reviewer VinQ. In short, we mainly think this has to do with **the nature of code errors and the strictness of the pass@1 metric**; even one small change or syntax error can result in a failure for code to execute properly, making it more brittle.
>
> ## Scaling experiments
> We thank the reviewer for this thoughtful suggestion. **Our paper does include experiments at both 1B and 3B for all variants** (1B results in Appendix F, Figures 7–8, Tables 2–6), though we notice that the comparison across scales was not easy to find. We consolidate the key patterns below.
>
> **Likelihood-Based Evaluations (1B→3B)**
>
> | Benchmark | Scale | BLT | BLT-D-4 | BLT-D-8 | BLT-D-16 |
> |---|---|---|---|---|---|
> | ARC-Easy | 1B | 63.21 | 60.76 | 61.06 | 59.83 |
> | | 3B | 74.33 | 72.39 | 70.95 | 66.89 |
> | | **% increase** | **+17.6%** | **+19.1%** | **+16.2%** | **+11.8%** |
> | ARC-Challenge | 1B | 34.94 | 32.96 | 34.16 | 32.88 |
> | | 3B | 45.75 | 41.46 | 41.03 | 40.43 |
> | | **% increase** | **+30.9%** | **+25.8%** | **+20.1%** | **+23.0%** |
> | PIQA | 1B | 75.46 | 74.48 | 73.56 | 72.36 |
> | | 3B | 79.38 | 79.60 | 78.02 | 76.93 |
> | | **% increase** | **+5.2%** | **+6.9%** | **+6.1%** | **+6.3%** |
> | HellaSwag | 1B | 60.17 | 59.06 | 58.34 | 57.13 |
> | | 3B | 74.98 | 71.86 | 70.56 | 69.12 |
> | | **% increase** | **+24.6%** | **+21.7%** | **+21.0%** | **+21.0%** |
> | MMLU | 1B | 33.90 | 33.60 | 32.28 | 32.09 |
> | | 3B | 41.15 | 39.07 | 38.29 | 37.08 |
> | | **% increase** | **+21.4%** | **+16.3%** | **+18.6%** | **+15.6%** |
>
> **Generation Tasks (BLT-D α=0.7, BLT-DV one-step; 1B→3B)**
>
> | Task | Scale | BLT | BLT-D-4 | BLT-DV-4 | BLT-D-8 | BLT-DV-8 | BLT-D-16 | BLT-DV-16 |
> |---|---|---|---|---|---|---|---|---|
> | FR→EN (BLEU) | 1B | 33.08 | 30.53 | 32.65 | 28.32 | 30.80 | 25.55 | 27.84 |
> | | 3B | 40.72 | 38.09 | 38.89 | 37.09 | 38.66 | 34.05 | 35.23 |
> | | **% increase** | **+23.1%** | **+24.8%** | **+19.1%** | **+31.0%** | **+25.5%** | **+33.3%** | **+26.5%** |
> | DE→EN (BLEU) | 1B | 31.46 | 27.73 | 29.56 | 25.20 | 27.71 | 23.19 | 24.96 |
> | | 3B | 38.82 | 36.29 | 37.46 | 35.29 | 37.11 | 32.62 | 34.52 |
> | | **% increase** | **+23.4%** | **+30.9%** | **+26.7%** | **+40.0%** | **+33.9%** | **+40.7%** | **+38.3%** |
> | HumanEval (pass@1) | 1B | 12.80 | 9.76 | 9.15 | 6.71 | 7.93 | 5.49 | 8.54 |
> | | 3B | 22.56 | 18.90 | 18.90 | 15.85 | 16.46 | 9.76 | 14.02 |
> | | **% increase** | **+76.3%** | **+93.6%** | **+106.6%** | **+136.2%** | **+107.6%** | **+77.8%** | **+64.2%** |
> | MBPP (pass@1) | 1B | 14.00 | 12.60 | 13.80 | 8.60 | 10.80 | 8.20 | 9.20 |
> | | 3B | 29.60 | 26.00 | 27.20 | 20.80 | 27.00 | 15.80 | 19.00 |
> | | **% increase** | **+111.4%** | **+106.3%** | **+97.1%** | **+141.9%** | **+150.0%** | **+92.7%** | **+106.5%** |
>
> The key patterns are consistent across both types of evaluations. **All variants (BLT, BLT-D, and BLT-DV) benefit substantially from scaling 1B→3B.** Coding tasks gain 64–150% versus 19–41% for translation and 5–31% for likelihood benchmarks. Diffusion-only variants (BLT-D) generally see the largest relative gains from 1B to 3B. Larger block sizes (BLT-D-16) still trail in absolute quality at both scales, but scaling and verification together substantially mitigate the speed-quality tradeoff of block diffusion.
>
> While we consolidate performance info here, we would also like to note that memory bandwidth reductions remain stable across scales (e.g., BLT-D-4 on MBPP: 70.1% at 1B vs 70.9% at 3B). Notably, BLT-S's lossless speedups actually improve at 3B: at k=16 on FR→EN, memory bandwidth reduction grows from 48.7% to 62.2%. These results are in the appendices of the paper.
>
> While a full scaling law study would be valuable future work, these results provide meaningful evidence that our methods' efficiency-quality trade-offs are preserved across scales.

---

> > ### Author Rebuttal · Reviewer_ErPH · 2026-04-01
> >
> > Thanks for providing the response. I would like to keep my positive score.
> >
> > More comments on the scaling experiment (W3):
> > - Including models at 1B and 3B can show the scaling effect to some extent. From existing results in the paper, it seems that the proposed method has good scaling features.
> > - However, providing results only at 2 scales is *not* a scaling-law experiment. I recommend running experiments at *multiple* scales and provide the hyperparameters in the scaling law to show that the scaling feature is *actually* better. This is commonly conducted by training extremely small models, e.g., ~100M models,  to obtain the scaling low.
> > - I acknowledge that (1) 1B and 3B results are included in the paper; (2) the margin of performance gain of BLT increases on larger models. Therefore, I understand that it is (1) not feasible for the authors to conduct large scale scaling experiments during rebuttal period; (2) small scale scaling experiments might not be helpful as BLT underperforms Transformer when the model sizes are smaller.
> > - I appreciate authors' explanation on this topic, and I totally accept the response to W3 now. However, exploring this direction would strongly enhance the soundness of this paper.

---

> > > ### Author Response · Authors · 2026-04-06
> > >
> > > We thank Reviewer ErPH for their thoughtful engagement during the rebuttal period! To address their concerns regarding scaling, we trained additional 400M-parameter BLT and BLT-D models from scratch, enabling us to study scaling behavior across three model sizes (400M, 1B, 3B). We use negative log-likelihood (NLL) on HellaSwag to fit scaling laws, and we report additional downstream evaluations below.
> > >
> > > ## Scaling Law on HellaSwag NLL
> > >
> > > All models were trained on 1T tokens. We fit single-variable power-law scaling curves of the form $L(N) = A \cdot N^{-\alpha} + L_\infty$ to HellaSwag NLL across three scales (400M, 1B, 3B). Results are summarized in the table below.
> > >
> > > | **Model** | $\alpha$ | $A$ | $L_\infty$ | $R^2$ | Pred. 7B | Pred. 13B |
> > > |-------|----------|-------------|------------|-------|----------|-----------|
> > > | **BLT-D-4** | 0.0204 | 1.9686 | 0.0000 | 0.9780 | 1.2398 | 1.2243 |
> > > | **BLT-D-8** | 0.0222 | 2.0123 | 0.0235 | 0.9989 | 1.2405 | 1.2239 |
> > > | **BLT-D-16** | 0.0209 | 1.9676 | 0.0238 | 0.9986 | 1.2480 | 1.2323 |
> > > | **BLT** | 0.0227 | 2.0594 | 0.0000 | 0.9689 | 1.2306 | 1.2135 |
> > >
> > > We also provide a log-log plot of these curves and encourage the reviewer to examine it: [[plot]](https://anonymous.4open.science/r/fast-blt-rebuttal-620C/scaling-laws.png)
> > >
> > > All variants yield strong fits ($R^2 \geq 0.97$). The scaling exponents are comparable across variants ($\alpha \approx 0.020$–$0.023$), indicating that **all architectures scale at similar rates**. Notably, BLT consistently achieves the lowest projected loss (1.2306 at 7B, 1.2135 at 13B), followed closely by BLT-D-4 and BLT-D-8, with BLT-D-16 trailing slightly. This suggests that BLT maintains a persistent offset advantage that is expected to hold at larger scales.
> > >
> > > We acknowledge several limitations. The fits are underconstrained (three data points for three free parameters) so we emphasize the relative ordering across variants rather than the absolute predictions. More broadly, as the reviewer notes, small-scale experiments carry inherent uncertainty, and the scaling behavior of diffusion language models remains underexplored. In particular, diffusion LMs may be more data-hungry than autoregressive models, meaning that scaling along the token/data axis could affect the relative comparison differently than scaling parameters alone. We were unable to investigate larger scales (e.g., 8B+) or varied data budgets within the rebuttal period, but plan to add models at additional scales in the final version to strengthen these conclusions.
> > >
> > > ## Additional Evaluations at 400M Scale
> > >
> > > We evaluated all models on translation tasks and likelihood-based benchmarks. We omitted coding benchmarks, as even the 400M BLT baseline cannot perform meaningfully on them. We’ve plotted these results, and we encourage the reviewer to examine them: [[translation]](https://anonymous.4open.science/r/fast-blt-rebuttal-620C/translation.png) [[likelihood]](https://anonymous.4open.science/r/fast-blt-rebuttal-620C/likelihood.png)
> > >
> > > First, we note that **all model variants (BLT, BLT-D, and BLT-DV at every block size) improve consistently with scale across both evaluation types.** In particular, for the translation tasks, **the gap between BLT and BLT-D variants consistently narrows with scale**, and models with larger diffusion block sizes benefit more from additional parameters. On likelihood-based benchmarks, the gap narrows from 400M to 1B but widens again from 1B to 3B. However, we believe the translation results are particularly informative, as they exercise the generative capabilities of the models, the primary mode of use at inference time for diffusion.
> > >
> > > ## Summary
> > >
> > > During this rebuttal, we conducted the following additional experiments:
> > >
> > > 1. **Pre-trained 400M-parameter versions of four model variants from scratch.**
> > > 2. **Fit scaling laws across three scales for all four variants.**
> > > 3. **Ran seven downstream task evaluations spanning three scales and seven model/generation variants.**
> > >
> > > We hope these experiments help clarify the scaling properties of our approach, and we kindly ask the reviewer to consider raising their score.

---

### Official Review · Reviewer_pS5W · 2026-03-05

**Soundness:** 1
**Presentation:** 3
**Significance:** 3
**Originality:** 3
**Overall Recommendation:** 4
**Confidence:** 3

**Summary:**

This paper introduces training and inference extensions for BLT to accelerate byte-level language model inference. The authors propose BLT-D, which trains the local decoder with an auxiliary block-wise masked diffusion objective to generate multiple bytes in parallel.
Additionally, the paper introduces BLT-S and BLT-DV, which leverage the model's lightweight local decoder to draft candidate bytes that are subsequently verified. Empirical results on 1B and 3B parameter models demonstrate substantial reductions in theoretical memory bandwidth and NFEs across translation and coding tasks.

**Compliance With Llm Reviewing Policy:**

Affirmed.

**Final Justification:**

During the rebuttal, the author provides wall-clock speedup results, which are helpful to validate the practical deployment. I also agree with the positive feedback from other reviewers. So I increase my score to positive.

**Key Questions For Authors:**

Can you provide actual wall-clock latency measurements (e.g., bytes/sec) on GPUs to validate that the theoretical NFE and memory bandwidth reductions translate to real-world speedups?

How does the end-to-end wall-clock latency and task performance of BLT-D/BLT-DV compare against a standard Transformer with a tokenizer of equivalent parameter count?

**Limitations:**

yes

**Strengths And Weaknesses:**

Strengths:

Originality: This work combines block-wise discrete diffusion and self-speculative decoding into BLT, which is an intuitive and novel contribution.

Significance: Addressing the memory bandwidth and inference latency bottlenecks of BLT is a highly important problem for making these models practically deployable.

Presentation: The paper is well-structured, with clear diagrams illustrating the attention masking patterns.

Weaknesses:

Soundness: The evaluation relies entirely on theoretical proxy metrics (NFEs and estimated memory bandwidth) rather than actual wall-clock speedups, completely ignoring the real-world CUDA/hardware overhead of the diffusion unmasking and verification steps. I wonder whether a real speedup can be achieved; this is my main concern with the paper. If the authors can justify this or provide more empirical results, I will consider updating my score.

---

> ### Author Rebuttal · Authors · 2026-03-30
>
> The authors would like to thank Reviewer pS5W for their thoughtful review!
>
> ## Wall-clock measurements and NFEs as an evaluation metric
> We would first like to note that **NFEs are a standard proxy metric in the discrete diffusion literature because they isolate algorithmic efficiency from implementation-specific factors.** This convention is followed in prominent works including SEDD (ICML 2024 Best Paper) and Block Diffusion (ICLR 2025 Oral), neither of which reports wall-clock time. Our contribution is a new BLT architecture and extensions that achieve comparable task performance with substantially fewer NFEs, a result that holds independently of any particular hardware or implementation.
>
> That said, we recognize the importance of runtime experiments, and we have conducted additional benchmarking with our current implementation to address this concern.
>
> ## Runtime estimation methodology and results
> To estimate end-to-end latency, **we benchmarked the forward-pass latency of each model component** (encoder, global Transformer, decoder) on H200 GPUs, using torch.cuda.synchronize() at all timing boundaries. We discarded the first 5 passes as warm-up and computed the mean over 10 subsequent passes. Total runtime was extrapolated by multiplying each component's mean per-pass latency by the corresponding number of NFEs.
>
> ### FR→EN (BLEU)
> | Model | Generation Setting | Task Score | Sampling Strategy | Est. Runtime (s) | Runtime Decrease vs. BLT (%) |
> |---|---|---|---|---|---|
> | BLT 3B | BLT (AR) | 40.72 | — | 21.06 | — |
> | BLT 3B | BLT-S (k=4) | 40.72 | — | 16.89 | 19.80 |
> | BLT 3B | BLT-S (k=8) | 40.72 | — | 13.41 | 36.32 |
> | BLT 3B | BLT-S (k=16) | 40.72 | — | 13.14 | 37.61 |
> | BLT-D-4 3B | BLT-D | 38.09 | Confidence, α=0.7 | 8.79 | 58.26 |
> | BLT-D-4 3B | BLT-DV | 38.89 | one step | 13.06 | 38.01 |
> | BLT-D-8 3B | BLT-D | 37.09 | Confidence, α=0.7 | 5.28 | 74.93 |
> | BLT-D-8 3B | BLT-DV | 38.66 | one step | 8.00 | 62.01 |
> | BLT-D-16 3B | BLT-D | 34.05 | Confidence, α=0.7 | 3.54 | 83.19 |
> | BLT-D-16 3B | BLT-DV | 35.23 | one step | 5.96 | 71.72 |
>
> ### DE→EN (BLEU)
> | Model | Generation Setting | Task Score | Sampling Strategy | Est. Runtime (s) | Runtime Decrease vs. BLT (%) |
> |---|---|---|---|---|---|
> | BLT 3B | BLT (AR) | 38.82 | — | 19.87 | — |
> | BLT 3B | BLT-S (k=4) | 38.82 | — | 16.81 | 15.40 |
> | BLT 3B | BLT-S (k=8) | 38.82 | — | 13.31 | 33.01 |
> | BLT 3B | BLT-S (k=16) | 38.82 | — | 13.06 | 34.29 |
> | BLT-D-4 3B | BLT-D | 36.29 | Confidence, α=0.7 | 8.77 | 55.87 |
> | BLT-D-4 3B | BLT-DV | 37.46 | one step | 12.94 | 34.90 |
> | BLT-D-8 3B | BLT-D | 35.29 | Confidence, α=0.7 | 5.29 | 73.37 |
> | BLT-D-8 3B | BLT-DV | 37.11 | one step | 8.24 | 58.51 |
> | BLT-D-16 3B | BLT-D | 32.62 | Confidence, α=0.7 | 3.22 | 83.81 |
> | BLT-D-16 3B | BLT-DV | 34.52 | one step | 5.29 | 73.35 |
>
> ### HumanEval (PASS@1)
> | Model | Generation Setting | Task Score | Sampling Strategy | Est. Runtime (s) | Runtime Decrease vs. BLT (%) |
> |---|---|---|---|---|---|
> | BLT 3B | BLT (AR) | 22.56 | — | 18.29 | — |
> | BLT 3B | BLT-S (k=4) | 22.56 | — | 16.36 | 10.56 |
> | BLT 3B | BLT-S (k=8) | 22.56 | — | 12.33 | 32.58 |
> | BLT 3B | BLT-S (k=16) | 22.56 | — | 10.55 | 42.31 |
> | BLT-D-4 3B | BLT-D | 18.90 | Confidence, α=0.7 | 8.09 | 55.80 |
> | BLT-D-4 3B | BLT-DV | 18.90 | one step | 12.57 | 31.26 |
> | BLT-D-8 3B | BLT-D | 15.85 | Confidence, α=0.7 | 4.37 | 76.10 |
> | BLT-D-8 3B | BLT-DV | 16.46 | one step | 7.28 | 60.20 |
> | BLT-D-16 3B | BLT-D | 9.76 | Confidence, α=0.7 | 2.57 | 85.95 |
> | BLT-D-16 3B | BLT-DV | 14.02 | one step | 4.51 | 75.33 |
>
> ### MBPP (PASS@1)
> | Model | Generation Setting | Task Score | Sampling Strategy | Est. Runtime (s) | Runtime Decrease vs. BLT (%) |
> |---|---|---|---|---|---|
> | BLT 3B | BLT (AR) | 29.60 | — | 13.78 | — |
> | BLT 3B | BLT-S (k=4) | 29.60 | — | 8.24 | 40.19 |
> | BLT 3B | BLT-S (k=8) | 29.60 | — | 6.25 | 54.63 |
> | BLT 3B | BLT-S (k=16) | 29.60 | — | 5.52 | 59.97 |
> | BLT-D-4 3B | BLT-D | 26.00 | Confidence, α=0.7 | 4.16 | 69.80 |
> | BLT-D-4 3B | BLT-DV | 27.20 | one step | 6.32 | 54.15 |
> | BLT-D-8 3B | BLT-D | 20.80 | Confidence, α=0.7 | 2.31 | 83.23 |
> | BLT-D-8 3B | BLT-DV | 27.00 | one step | 3.73 | 72.93 |
> | BLT-D-16 3B | BLT-D | 15.80 | Confidence, α=0.7 | 1.45 | 89.50 |
> | BLT-D-16 3B | BLT-DV | 19.00 | one step | 2.53 | 81.66 |
>
> **BLT-D, BLT-DV, and BLT-S all achieve substantial runtime reductions over BLT.** These estimates only reflect per-component GPU compute cost; together with our memory bandwidth analysis, they provide a more holistic picture of the efficiency gains. We thank the reviewer for suggesting these experiments.
>
> ## Comparison with a standard Transformer
> We agree this is important for showing the practicality of byte-level models; unfortunately, training a 3B subword model during the rebuttal period is not feasible for us. We refer to the BLT paper, which shows that BLT matches or outperforms parameter-matched Llama models on many tasks, which is why we found BLT to be the appropriate baseline.

---

> > ### Author Rebuttal · Reviewer_pS5W · 2026-04-02
> >
> > Thank you for your response. The wall-clock speedup is helpful. I think including a figure showing speed-accuracy trade-off of different models (BLT-D, BLT, convention LLM) in the future version of the paper would be helpful.
> >
> > I will consider increasing the score to positive.

---

### Official Review · Reviewer_VinQ · 2026-03-13

**Soundness:** 3
**Presentation:** 2
**Significance:** 2
**Originality:** 3
**Overall Recommendation:** 4
**Confidence:** 2

**Summary:**

This paper addresses the inference bottleneck of byte-level language models, specifically the Byte Latent Transformer (BLT). It proposes three methods: (1) BLT Diffusion (BLT-D), which adds an auxiliary block-wise masked diffusion objective during training so the decoder can generate multiple bytes in parallel at inference time; (2) BLT Self-speculation (BLT-S), where BLT's lightweight decoder speculatively generates bytes beyond its normal patch boundary, then verifies with a full forward pass; and (3) BLT Diffusion+Verification (BLT-DV), which uses diffusion to draft a block of bytes and then verifies them using BLT-D's autoregressive next-byte prediction capability. Experiments on 1B and 3B models across translation (FLORES) and coding (HumanEval, MBPP) tasks show 50-91% memory bandwidth reduction compared to BLT, with varying quality-speed tradeoffs.

**Compliance With Llm Reviewing Policy:**

Affirmed.

**Final Justification:**

My concerns are addressed and I will value the paper as the rating of 4.

**Key Questions For Authors:**

- Why does coding performance degrade much more than translation at larger block sizes? Is this related to the entropy structure of code vs. natural language?

- Have you measured the verification acceptance rates for BLT-DV across different tasks? If acceptance rates are low on code, this would explain the quality gap and suggest the method is task-dependent.

**Strengths And Weaknesses:**

## Strengths

- BLT-D is the genuinely novel contribution here. The idea of training a hierarchical byte-level model with a joint next-byte prediction + block-wise diffusion objective, where the diffusion blocks are aligned to BLT's variable-length patch structure but allowed to extend beyond patch boundaries, is non-trivial. The decoder attention masking (causal for clean prefix, bidirectional within masked blocks, causal from blocks to prefix) is carefully designed to be compatible with both training objectives. This is a real architectural contribution that required thought about how diffusion and dynamic patching interact.

- The experimental methodology is thorough and well-structured. The paper trains 8 models (4 types × 2 sizes), evaluates on 4 generation tasks + 5 likelihood benchmarks, sweeps over multiple unmasking strategies (confidence-based and entropy-bounded) and hyperparameters, and reports NFEs for each model component separately (decoder, encoder/global). The memory bandwidth metric (Eq. 8) is a reasonable proxy for actual latency in memory-bound settings, and the paper is upfront about when this metric applies (small batch, latency-oriented serving).

- The presentation is clear. The figures, especially Figure 1 (BLT-D inference), Figure 2 (training forward pass), and the attention mask visualizations, make the architecture easy to follow. The algorithms in the appendix are precise and complete.

## Weaknesses

- BLT-S is self-speculative decoding applied to BLT — the decoder generates beyond its normal boundary, then the full model verifies. This is exactly the Draft & Verify / self-speculative decoding paradigm adapted to BLT's encoder-decoder hierarchy. The paper claims it "fundamentally differs" because it uses BLT's existing lightweight decoder rather than a separate small model, but self-speculative decoding by definition uses the same model with a cheaper component for drafting. Layer-skipping approaches (e.g., LayerSkip) do the same thing. The only difference here is that BLT happens to have a natural cheap component (the decoder) — the algorithmic idea is not new.

- BLT-DV is diffusion-as-draft + autoregressive verification, which is the standard speculative decoding pattern with diffusion as the drafter. The paper even acknowledges that BLT-DV uses "the same procedure as in Algorithm 2" for verification. Recent concurrent work on block/semi-autoregressive diffusion models (Block Diffusion by Arriola et al. 2025, Set Block Decoding by Gat et al. 2025) has explored similar ideas of generating blocks then verifying/refining. The contribution is applying this to BLT specifically, but the conceptual advance is limited.

- The evaluation is narrow — only translation (FLORES) and code generation (HumanEval, MBPP). These are relatively short-output tasks. For a paper about inference efficiency, the absence of long-form generation benchmarks (summarization, open-ended QA, chat) is a significant gap. The efficiency gains may look very different on tasks with thousands of output bytes, where the amortized cost structure changes and the tradeoff between block size and quality degradation could be more severe.

- The quality degradation at larger block sizes is significant and underexplored. BLT-D-16 drops from 39.02→22.56 on HumanEval pass@1 (3B), which is a 42% relative decrease. On MBPP it goes from 31.80→24.00. The paper frames this as a "trade-off" but doesn't analyze why coding tasks degrade much more than translation. Is it because code has higher byte-level entropy? Because syntactic constraints make parallel generation harder? Understanding this failure mode is important for practical deployment.

---

> ### Author Rebuttal · Authors · 2026-03-30
>
> We thank Reviewer VinQ for the thoughtful review and for recognizing our thorough experiments and clear presentation!
>
> ## Novelty of BLT-S and BLT-DV
> We agree that BLT-S shares the draft-then-verify pattern with prior self-speculative methods, and will cite LayerSkip in the revision. However, BLT-S differs in important ways. First, no training modification is needed: LayerSkip and similar work require a special recipe to make early-exit drafting viable, whereas BLT's decoder is already an effective next-byte predictor by design. Second, the efficiency gains are mechanically different: for example, LayerSkip saves compute by doing less work per call, while BLT-S calls the decoder more often but the encoder and global model less often, which redistributes work to the lightweight component. More broadly, **we show that hierarchical architectures like BLT naturally produce a component well-suited for speculation, an important observation not previously identified in previous work** on BLT, H-Nets, Hourglass Transformers, etc.
>
> BLT-DV is conceptually related to Set Block Decoding and similar work, which we cite. Its value is showing that BLT-D's joint training objective naturally enables verification. The same attention masks used for autoregressive BLT inference are reused directly, meaningfully recovering quality lost by diffusion-only decoding. We will make this framing clearer in the revision.
>
> ## Evaluation breadth
> We agree that longer-output tasks are important. We have added **document-level DE→EN translation** results using the WMT 2018 news test set (347 documents, ~910 bytes average target length). Due to the rebuttal length limit, we report summary statistics for BLT-D-8 and BLT-D-16 variants here; full results (including acceptance rates) will appear in the final paper. We use confidence-based unmasking (α=0.7) for BLT-D, and one-step diffusion for BLT-DV.
>
> | Model | BLEU | Dec. NFEs | Enc/Glob NFEs | Memory Decrease vs. BLT (%) | Runtime Decrease vs. BLT (%) |
> |---|---|---|---|---|---|
> | BLT 3B | 33.99 | 2000 | 394 | — | — |
> | BLT-D-8 3B | 33.73 | 527 | 250 | 45% | 58% |
> | BLT-DV-8 3B | 35.51 | 500 | 500 | −4% | 31% |
> | BLT-D-16 3B | 32.14 | 465 | 125 | 70% | 73% |
> | BLT-DV-16 3B | 34.71 | 336 | 336 | 30% | 54% |
>
> **Efficiency gains hold on longer outputs:** BLT-D-8 achieves 58% runtime reduction while nearly matching BLT's BLEU, and BLT-D-16 achieves 73% reduction with little quality loss. BLT-DV-8 and BLT-DV-16 both outperform BLT while delivering substantial speedups. We refer to our response to Reviewer pS5W for runtime methodology details.
>
> More generally, our models are base models, so we report results on tasks that are appropriate for base models. We evaluated summarization, but found that even the baseline BLT model struggled with the task. Summarization and chat-oriented tasks are better suited to post-trained models; we have chosen not to focus on them because post-training is outside the scope of this work.
>
> ## Quality degradation at larger block sizes
> Following the reviewer's suggestion, we computed both marginal byte entropy (Shannon entropy over byte frequencies) and conditional byte entropy from BLT's pretrained entropy model (mean per-byte surprisal):
>
> | Dataset | N | Marginal (bits/byte) | Cond. Mean (bits/byte) | Cond. σ |
> |---|---|---|---|---|
> | Flores FR (source) | 1,012 | 4.27 | 1.54 | 0.25 |
> | Flores DE (source) | 1,012 | 4.33 | 1.73 | 0.27 |
> | Flores EN (targets) | 1,012 | 4.16 | 1.21 | 0.22 |
> | All translation text | 3,036 | 4.25 | 1.50 | 0.33 |
> | MBPP solutions (code) | 974 | 4.20 | 1.27 | 0.37 |
> | HumanEval solutions (code) | 164 | 3.76 | 1.23 | 0.59 |
> | All code (MBPP+HE) | 1,138 | 4.14 | 1.27 | 0.41 |
>
> Strikingly, **code is *more* predictable at the byte level on average**. However, code exhibits substantially higher *variance* in conditional entropy. So most bytes in code are very predictable, but the entropy pattern is spikier. Given that code is more predictable but still difficult for our models, this might suggest that certain bytes carry disproportionate importance, and even a single incorrect byte may break code execution. That makes pass@1 particularly harsh for code, since it rewards only a fully correct first attempt. **It is the variance in byte importance, compounded by the strictness of the pass@1 metric, that makes coarser block sizes challenging for code generation.**
>
> ## Verification acceptance rates
> The reviewer asks whether acceptance rates are lower for code, which is an excellent question. **Acceptance rates broken down by task are reported in Tables 3–10 in Appendices F and G of the paper.** **Code acceptance is actually *higher*, consistent with code's lower byte entropy.** This further supports the error tolerance explanation: the issue is not that code drafts are rejected more often, but that the errors may be influential to overall code correctness. Acceptance rates for the new document-level translation task will be provided in the final paper.

---

> > ### Author Rebuttal · Reviewer_VinQ · 2026-04-01
> >
> > Thank the authors for the rebuttal. I'll raise my rating to 4.

---

### Decision · Program_Chairs · 2026-04-30

**Decision:**

Accept (regular)

**Comment:**

The paper proposes three interesting methods to improve the inference costs of the Byte Latent Transformer. Reviewers felt the contribution is novel, the presentation clear, and the experimental methodology thorough and sound. The authors successfully addressed the specific questions and weaknesses posed, often supported by new experimental results, such as benchmarking wall clock speeds.